# Robust Representation Learning via Asymmetric Negative Contrasting and Reverse Attention

## Abstract

Deep neural networks are vulnerable to adversarial noise. Adversarial training (AT) has been demonstrated to be the most effective defense strategy to protect neural networks from being fooled. However, we find AT omits to learning robust features, resulting in poor performance of adversarial robustness. To address this issue, we highlight two characteristics of robust representation: *(1) exclusion: the feature of natural examples keeps away from that of other classes; (2) alignment: the feature of natural and corresponding adversarial examples is close to each other.* These motivate us to propose a generic framework of AT to gain robust representation, by the asymmetric negative contrast and reverse attention. Specifically, we design an asymmetric negative contrast based on predicted probabilities and generate adversarial negative examples by the targeted attack, to push away examples of different classes in the feature space. Moreover, we propose to weight feature by parameters of the linear classifier as the reverse attention, to obtain class-aware feature and pull close the feature of the same class. Empirical evaluations on three benchmark datasets show our methods greatly advance the robustness of AT and achieve the state-of-the-art performance.

## 1 Introduction

Deep neural networks (DNNs) have achieved great success in academia and industry, but they are easily fooled by carefully crafted adversarial examples to output incorrect results [13], which leads to potential threats and insecurity in application. Given a well-trained DNN and a natural example, an adversarial example can be generated by adding small perturbation that is invisible to the human eyes to the natural example. The natural example can be correctly classified before the perturbation and the adversarial example is incorrectly classified after the perturbation. In recent years, there are many researches exploring the generation of adversarial examples to cheat models in various fields, including image classification [13, 26, 5, 9], object detection [33, 8], natural language processing [27, 2], semantic segmentation [28, 25], etc. The vulnerability of DNNs has aroused common concerns on adversarial robustness.

Many empirical defense methods have been proposed to protect DNNs from adversarial perturbation, such as adversarial training (AT) [26, 36, 30, 18, 39, 37, 31], image denoising [24], defensive distillation [38, 6] and so on. The mainstream view is that AT is the most effective defense, which has a training process of a two-sided game. The "attacker" crafts perturbation dynamically to generate adversarial data to cheat the "defender", and the "defender" minimizes the loss function against adversarial samples to improve robustness of models. Existing work [38, 6, 37, 11, 18, 20, 39] has improved the effectiveness of AT in many aspects, but few studies pay attention to learning robust feature. The overlook may lead to potential threats in the feature space of AT models, which does harm to robust classification. Besides, there are no criteria for robust feature. In addition, adversarial

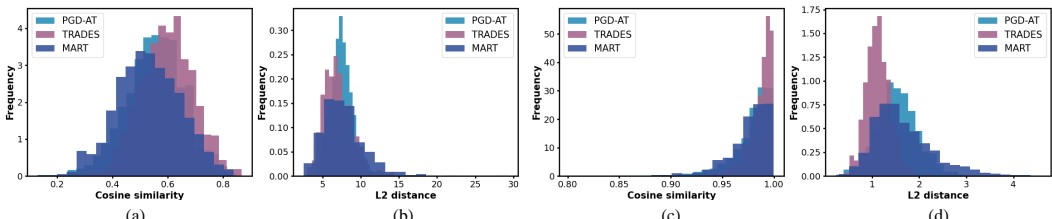

(a)          (b)          (c)          (d)

Figure 1: Frequency histograms of the $L_2$ distance and cosine similarity of the feature that belongs to natural examples, AEs and OEs. The four figures show the cosine similarity of the feature between natural examples and OEs (a), the $L_2$ distance of the feature between natural examples and OEs (b), the cosine similarity of the feature between natural examples and AEs (c), the $L_2$ distance of the feature between natural examples and AEs (d), respectively. The feature denotes the feature vector $z$ before the linear layer. We train ResNet-18 [15] models on CIFAR-10 [22] with three AT methods: PDG-AT [26], TRADES [36] and MART [30]. In the calculation, we use all samples labeled as class 0 in the test set as natural examples and generate AEs by PGD-10 [26].

contrastive learning (ACL) and robust feature selection (RFS) are techniques to optimize feature distribution. ACL [21, 12, 35] is a kind of contrast learning (CL) [7, 17, 14] that extends to AT. RFS mostly modifies the architecture of models [32, 1, 34] to select important feature. However, the target problems of them are not to learn robust feature.

To demonstrate AT is indeed deficient in the representation which causes limited adversarial robustness, we conduct a simple experiment. We choose the $L_2$ distance and cosine similarity as metrics. And we measure the distance and similarity of the feature between natural examples, adversarial examples (AEs) and examples of other classes (OEs). The frequency histograms of the distance and similarity is shown in Figure 1. Figure 1 (a) and Figure 1 (b) show that the cosine similarity of the feature between natural examples and OEs shows a Gaussian distribution between 0.4 and 0.8, and the $L_2$ distance shows a skewed distribution between 2.0 and 12.0, which indicates there are very close pairs of natural examples and OEs that are not distinguished in the feature space. In Figure 1 (c) and Figure 1 (d), it is shown that there are a skewed distribution between 0.9 and 0.99 for the cosine similarity of the feature between natural examples and AEs, and a skewed distribution between 0.5 and 2.5 for the $L_2$ distance, which indicates that the feature of natural examples and AEs is not adequately aligned. Thus, there is still large room for optimization of the feature of AT.

Based on the observation, we propose two characteristics of robust feature: *exclusion: the feature of natural examples keeps away from that of other classes; alignment: the feature of natural and corresponding adversarial samples is close to each other*. First, *exclusion* confirms the separability between different classes and avoids confusion in the feature space, which makes it hard to fool the model because the feature of different classes keep a large distance. Second, *alignment* insures the feature of natural examples is aligned with adversarial one, which guarantees the predicted results of the natural and adversarial examples of the same instances are also highly consistent. And it helps to narrow the gap between robust accuracy and clean accuracy.

To address the issue, we propose an AT framework to concentrate on robust representation with the guidance of the two characteristics. Specifically, we suggest two strategies to meet the characteristics, respectively. Treat a natural example and corresponding AE as a positive pair (PP), and treat a natural example and corresponding OE as a negative pair (PP). For *exclusion*, we propose an asymmetric negative contrast based on predicted probabilities, which freezes natural examples and pushes away OEs by reducing the confidence of predicted class when predicted classes of NPs are consistent. In particular, we find OEs generated by the targeted attack are more beneficial for correct classification than those selected carefully. For *alignment*, we use the reverse attention to weight the feature of PPs by partial parameters of the linear classifier, which contains the importance of feature to target classes during classification. Because the feature of the same class gets the same weighting and feature of different classes is weighted disparately, PPs are aligned and each example of PPs becomes close to each other in the feature space. Empirical evaluations show that AT methods combined with our framework can greatly enhance robustness, which means the neglect of learning robust feature is one of the main reasons for poor robust performance of AT. In a word, we propose a generic AT framework with the Asymmetric Negative Contrast and Reverse Attention (**ANCRA**), to learn robust representation and advance robustness. Our main contributions are summarized as follows:

- We suggest improving adversarial training from the perspective of learning robust feature, and two characteristics are highlighted as criteria of optimizing robust representation.

- We propose a generic framework of adversarial training, termed as ANCRA, to obtain robust feature by the asymmetric negative contrast and reverse attention, with the guidance of two characteristics of robust feature. It can be easily combined with other defense methods.

- Empirical evaluations show our framework can obtain robust feature and greatly improve adversarial robustness, which achieves the of state-of-the-art performances on CIFAR-10, CIFAR-100 and Tiny-ImageNet.

## 2 Related work

**Adversarial training** Madry et al. [26] propose PGD attack and PGD-based adversarial training, forcing the model to correctly classify adversarial samples within the epsilon sphere during training to obtain robustness, which is the pioneer of adversarial learning. Zhang et al. [36] propose to learn both natural and adversarial samples and reduce the divergence of classification distribution of both to reduce the difference between robust accuracy and natural accuracy. Wang et al. [30] find that misclassified samples during training have a negative impact on robustness significantly, and propose to improve the model's attention to misclassification by adaptive weights. Zhang et al. [37] propose to replace fixed attack steps with attack steps that just cross the decision boundary, and improved the natural accuracy by appropriately reducing the number of attack iterations. Huang et al. [18] replace labels with soft labels predicted by the model and adaptively reduce the weight of misclassification loss to alleviate robust overfitting problem. Dong et al. [11] also propose a similar idea of softening label and explain the different effects of hard and soft labels on robustness by investigating the memory behavior of the model for random noisy labels. Chen et al. [6] propose random weight smoothing and self-training based on knowledge distillation, which greatly improve the natural and robust accuracy. Zhou et al. [39] embed a label transition matrix into models to infer natural labels from adversarial noise. However, little work has been done to improve AT from the perspective of robust feature learning. Our work shows AT indeed has defects in the feature distribution, and strategies proposed to learn robust feature can greatly advance robustness, which indicates the neglect of robust representation results in poor robust performance of AT.

**Adversarial contrastive learning** Kim et al. [21] propose an adversarial training method of maximizing and minimizing the contrastive loss. Fan et al. [12] notice that the robustness of ACL relies on fine-tuning, and pseudo labels and high-frequency information can advance robustness. Kucer et al. [23] find that the direct combination of self-supervised learning and AT penalizes non-robust accuracy. Bui et al. [3] propose some strategies to select positive and negative examples based on predicted classes and labels. Yu et al. [35] find the instance-level identity confusion problem brought by positive contrast and address it by asymmetric methods. The idea of these methods motivates us to further consider how to obtain robust feature by contrast mechanism. We design a new negative contrast to push away NPs and mitigate the confusion caused by negative contrast.

**Robust feature selection** Xiao et al. [32] take the maximum k feature values in each activation layer to increase adversarial robustness. Zoran et al. [40] use a spatial attention mechanism to identify important regions of the feature map. Bai et al. [1] propose to suppress redundant feature channels and dynamically activate feature channels with the parameters of additional components. Yan et al. [34] propose to amplify the top-k activated feature channels. Existing work has shown enlarging import feature channels is beneficial for robustness, but most approaches rely on extra model components and do not explain the reason. We proposes the reverse attention to weight feature by class information without any extra components, and explain it by *alignment* of feature.

## 3 Methodology

This section explains the instantiation of the our AT framework from the perspective of the two characteristics of robust feature. To meet *exclusion*, we design an asymmetric negative contrast based on predicted probabilities and propose to craft OEs by the targeted attack, to push away the feature of NPs. To confirm *alignment*, we propose the reverse attention to weight the feature of the same class, by the corresponding weight of target class in parameters of the linear classifier, so that the feature of PPs is aligned and the gap of the feature between natural examples and AEs becomes small.

### 3.1 Notations

In this paper, capital letters indicate random variables or vectors, while lowercase letters represent their realisations. We define the function for classification as $f(\cdot)$. It can be parameterized by DNNs. $Linear(\cdot)$ is the linear classifier with a weight of $\Omega$ (C, R), in which C denotes the class number and R denotes the channel number of the feature map. $g(\cdot)$ is the feature extractor, i.e., the rest model without $Linear(\cdot)$. Let $\mathcal{B} = \{x_i, y_i\}_i^N$ be a batch of natural samples where $x_i$ is labeled by $y_i$. Given an adversarial transformation $\mathcal{T}_a$ from an adversary $\mathcal{A}$ (e.g., PGD attack in [26]), and a strategy $\mathcal{T}_o$ for selection or generation of OEs. For data, we consider a positive pair PP=$\{x_i, x_i^a | x_i \in \mathcal{B}, x_i^a = \mathcal{T}_a(x_i)\}_i^N$, and a negative pair NP=$\{x_i, x_i^o | x_i \in \mathcal{B}, x_i^o = \mathcal{T}_o(x_i)\}_i^N$. Let $\mathbb{N}(x, \epsilon)$ represent the neighborhood of $x : \{\tilde{x} : \|\tilde{x} - x\| \leq \epsilon\}$, where $\epsilon$ is the perturbation budget. For an input $x_i$, we consider its feature $z_i$ before $Linear(\cdot)$, the probability vector $p_i = softmax(f(x_i))$ and predicted class $h_i = argmax(p_i)$, respectively.

### 3.2 Adversarial training with asymmetric negative contrast

Firstly, we promote AT to learn robust representation that meets *exclusion*. We notice that ACL has the contrastive loss [29] to maximize the consistency between PPs and to minimize the consistency between NPs. Motivated by the contrast mechanism, we consider to design a new negative-contrast term and combine it with AT loss, which creates a repulsive action between NPs when minimize the whole loss. Thus, we propose a generic pattern of AT loss with a negative contrast. Let TRADES [36] represent AT in the following paper as a example.

$$\mathcal{L}^{\text{CAL}}(x, y, x^a, x^o) = \mathcal{L}^{\text{TRADES}} + \text{Sim}(x, x^o) = \mathcal{L}_{CE}(x, y) + \mathcal{D}_{KL}(x, x^a) + \text{Sim}(x, x^o), \quad (1)$$

Where $x$ denotes natural examples with labels $y$, $x^a$ are AEs generated by untargeted PGD [26], $x^o$ are negative examples of other classes (OEs), $Sim$ is a similarity function, $\mathcal{L}_{CE}$ denotes the cross-entropy loss and $\mathcal{D}_{KL}$ denotes divergence of Kullback-Leibler. AEs generated by maximizing $\mathcal{L}_{CE}$ typically have wrong predicted classes, given by:

$$x_{t+1}^a := \prod_{\mathbb{N}(x, \epsilon)} \left( x_t^a + \epsilon \, \text{sign} \left( \nabla_x \mathcal{L}_{CE} \left( (f(x_t^a), y) \right) \right) \right), \quad (2)$$

where $\epsilon$ denotes the $L_\infty$-norm of perturbation, $x_t^a$ denotes adversarial positive samples after the $t$th attack iteration, $\mathbf{\Pi}$ denotes a clamp function, $sign$ denotes a sign function and $\nabla_x \mathcal{L}_{CE}$ denotes the gradient of $\mathcal{L}_{CE}$ with respect to $x$. When minimizing the loss in Eq 1, $\mathcal{L}^{\text{TRADES}}$ learns to classify natural examples and AEs correctly, and additional negative contrast prompts the inconsistency of NPs, which keeps the feature of NPs far away from each other. The whole loss guides the model to learn correct classification from TRADES and push away NPs from each other to ensure *exclusion*. Although we have a generic pattern of AT loss with a negative contrast, there are several problems about details to address. To refine the negative contrast and address problems, we further propose a method to calculate the negative contrast and strategy to generate OEs.

#### 3.2.1 Asymmetric negative contrast based on probabilities

The work in [35] has indicated that when the predicted classes of the adversarial positive examples (i.e., AEs) and negative samples (i.e., OEs) are the same, the positive contrast may lead to a conflict between the positive and negative contrast, resulting in wrong classification. On the basis, we find a similar conflict can also be caused by the negative contrast when the predicted classes of AEs and OEs are different, which we named by class confusion. As shown in Figure 2, when AEs and OEs have different predicted classes, natural examples are subject to the attraction of AEs and the repulsion of OEs at the same time. And it is likely to move near the decision boundary or even into the wrong class space under the actions, which does harm to *exclusion*.

In order to alleviate the problem of class confusion, We should reasonably control the effect of the repulsion of negative contrast between natural examples and OEs. we propose an asymmetric method of the negative contrast, $\text{Sim}^\alpha(x, x^o)$, to decouple the repulsive force into an one-side push from the natural example to the OE and an one-side push from the OE to the natural example, given by:

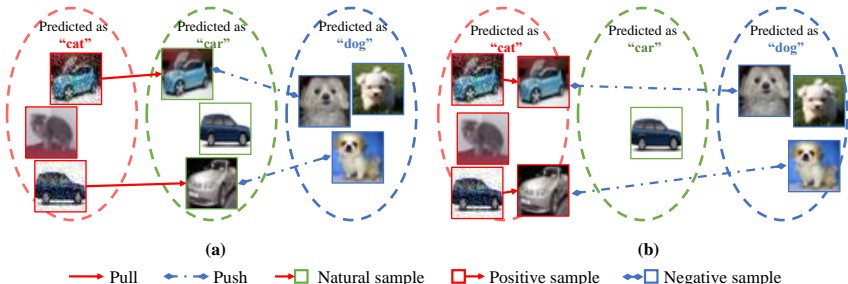

Figure 2: Illustrations of class confusion when the classes of positive examples (i.e., AEs) and negative examples (i.e., OEs) are different. (a) shows the normal situation before the optimization. (b) shows the situation of class confusion after the optimization. In each circle, data points have the same predicted class. In (a), AEs locate in the wrong predicted class different from natural example and OEs. The TRADES loss narrow the gap of classification between natural examples and AEs, and thus AEs in the wrong class pull natural examples to move toward the wrong class and the negative contrast pushes natural examples to leave from the original class. With these actions, natural examples come to the decision boundary and even into the wrong class easily as (b) shows.

$$\text{Sim}^{\alpha}(x, x^o) = \alpha \cdot \overline{\text{Sim}}(x, x^o) + (1 - \alpha) \cdot \overline{\text{Sim}}(x^o, x), \tag{3}$$

where $\overline{\text{Sim}}(x, x^o)$ denotes the one-sided similarity of $x$ and $x^o$. When minimizing $\overline{\text{Sim}}(x, x^o)$, we stop the back-propagation gradient of $x$ and only move $x^o$ away from $x$. $\alpha$ denotes the weighting factor to adjust the magnitude of the two repulsive forces. When $\alpha = 0$, OEs are frozen and only the feature of natural samples is optimized to push far away from the feature of OEs. As $\alpha$ increases, the natural sample becomes more repulsive to the OE and the OE pushes the natural example less. To mitigate the class confusion problem, we should choose $\alpha$ that tends to 1 to reduce the repulsive force from the OE to the natural example, to prevent the natural example from being pushed into the wrong class. Experiments show that $\alpha = 1$ leads to the best performance provided in our supplementary material), which pushes away NPs by only pushing off OEs and follows what we have expected.

Then we propose the negative contrast based on predicted probabilities, $\text{Sim}^{\alpha}_{cc}(x, x^o)$, to measure the repulsive force of NPs pushing away from each other. It pushes away NPs by decreasing the corresponding probabilities of the predicted classes when the predicted classes of NPs are consistent.

$$\text{Sim}^{\alpha}_{cc}(x, x^o) = \frac{1}{\|\mathcal{B}_i\|} \sum_{i=1}^{n} \mathbb{I}(h_i = h_i^o) \cdot \left[ \alpha \sqrt{\hat{p}_i(h_i) \cdot p_i^o(h_i)} + (1 - \alpha) \sqrt{p_i(h_i) \cdot \hat{p}_i^o(h_i)} \right], \tag{4}$$

where $\|\mathcal{B}_i\|$ denotes the batch size, $\mathbb{I}(\cdot)$ denotes the Indicator function and $\hat{p}$ denotes freezing the back-propagation gradient of $p$. $h_i$ and $h_i^n$ denote the predicted classes of the NP. And $p_i$ and $p_i^n$ denote the probability vectors of the NP. Under the negative contrast, the model pushes the natural example in the direction away from the predicted class of the OE and push the OE in the direction away from the predicted class of the natural example when and only when two predicted classes of the NP are consistent. This ensures that the action of *exclusion* not only pushes away the feature of NPs in the feature space, but also reduces the probabilities of NPs in the incorrect class. Since the negative contrast has only directions to reduce the confidence and no explicit directions to increase the confidence, it does not create any actions to push the natural example into the feature space of wrong classes even in the scenario of class confusion, which can effectively alleviate the problem.

### 3.2.2 Generate negative samples by targeted attack

To obtain OEs, previous negative sampling strategies [19] simply screen natural samples and pick up the negatives from them, but rarely consider generating special negative samples to assist learning. We innovatively propose a strategy to craft OEs by the targeted attack: natural negative examples with labels that is different from those of natural examples are attacked to the labeled classes of natural examples by targeted PGD-10 [26], to manufacture hard negatives containing adversarial noise.

$$x_{t+1}^o := \prod_{\mathbb{N}(x^o, \epsilon)} \left( x_t^o - \epsilon \, \text{sign} \left( \nabla_{x^o} \mathcal{L}_{CE} \left( (f(x_t^o), y) \right) \right) \right), \tag{5}$$

Where $\nabla_{x^o}\mathcal{L}_{CE}$ denotes the gradient of $\mathcal{L}_{CE}$ with respect to $x^o$. By this strategy, clean OEs randomly chosen from other classes are attacked to the labeled classes of natural examples and become negative adversarial examples. The motivation makes intuitive sense. 1) The negative adversarial sample generated by the targeted attack will be classified as the labeled class of the natural example with high confidence, but its ground truth label is not that, which makes it a very hard negative sample and is beneficial for the negative contrast. 2) The negative adversarial sample contains adversarial noise, which is special feature that natural negative samples do not have. And this feature helps the model learn the paradigm of adversarial noise and improve the robust performance. In particular, we demonstrate that negative samples with adversarial noise do improve robustness better in Table 4.

### 3.3 Adversarial training with reverse Attention

Secondly, we continue to improve TRADES to learn robust representation that meets *alignment*. Consider the calculating process of the model $f(\cdot)$. First, the feature vector $z$ is obtained by $g(x)$, and then the output vector $\Omega z$ is obtained by a linear mapping $Linear(z)$. Each element $z_i$ in $z$ represents the activation level of the feature channel that may be helpful for classification, with larger values representing more feature information extracted from that channel; $\omega^{i,j}$ in $\Omega$ represents the importance of the $i$th feature channel to the $j$th class, with higher values representing the greater contribution of the feature channel to the class. Motivated by [1, 34], we exploit the importance of feature channels to target classes to align the feature of examples of the same classes and pull close the feature of PPs, which is named by reverse attention. To be specific, we take the Hadamard product (Kronecker product) of partial weight of the classifier $\Omega^j$ and the feature vector $z$. It can weight feature channel by channel according to its contribution to being classified as the target class $j$, and gain a class-aware feature vector $z'$ containing the information of the target class $j$.

$$z'_i = \begin{cases} z_i \odot \omega^{i,y}, & \text{(training phase)} \\ z_i \odot \omega^{i,h(x)}, & \text{(testing phase)} \end{cases} \tag{6}$$

where $\odot$ denotes the Hadamard product operation, which is the method of multiplying two matrices of the same size element by element to obtain a new matrix of the same size. To ensure parameters used for weighting have the correct feature-to-class importance, we use the unweighted feature vector $z$ to go through $Linear(\cdot)$ to obtain the auxiliary probability vector $p$, and $z'$ to get the final probability vector $p'$. Finally, we use both $p$ and $p'$ to train the model. During the training phase, we use the true label $y$ as an indicator to determine the importance of channels, i.e., $\Omega^j = \Omega^y$. And in the testing phase, since the true label is not available, we simply choose a sub-vector of the linear weight by the predicted class $h(x)$ as the importance of channels. We add the reverse attention to the last feature layer in the model, which generally contains two blocks. The model with the reverse attention does not need any extra modules, but module interactions are changed.

Let' s make a detailed analysis and explanation of the principle of this method. The class information from labels guides the input image to be mapped from the feature to the classification vector during training, establishing an feature-to-class mapping relationship. In the model, the feature extractor captures the representation that is helpful for classification until the feature vector contains enough information that allows the classifier to classify the sample as the target class. Among all the modules, the classifier is the closest to labels and learns which feature channel plays an important role in being classified as the target class (i.e., the feature importance). Since the classifier is unique, the importance of the feature channels of one example is exactly the same with that of the other samples in the same class, benefiting the generalization and robustness of the model in the target class. We propose the reverse attention to utilize this information to improve feature rather than classification. The feature vectors are weighted by partial parameters of the linear layer that belong to the target class, which can change the activation of each channel adaptively according to the feature importance, acting as an attention with the guidance of the class information. After the attention, the important channels in the feature vector are boosted and the redundant channels are weakened, i.e., the information contributes to the target class will become larger and more significant, which is helpful for correct classification. Considering from the perspective of the feature distribution, the weighted feature has gained extra class information, which induces changes in the feature distribution. Feature vectors with the same target class get the same weighting, and thus the weighted feature becomes more similar. Moreover, feature vectors with different target classes are weighted according to different weights, and the weighted feature distributions become more inconsistent. Therefore, the reverse attention guides

the *alignment* of the feature of the examples in the same class, pulling the feature of PPs closer and pushing the feature of NPs far away, which benefits *alignment* and drops by to promote *exclusion* and classification. Aligned feature has similar activations in every feature channel, which helps the model narrows the gap between feature of natural examples and AEs.

# 4   Experiments

To demonstrate the effectiveness of the proposed approach, we show feature distribution of trained models firstly. Then we evaluate our framework against white-box attacks and adaptive attacks, and make a compare with other defense methods. We conduct experiments across different datasets and models. Because our methods are compatible with existing AT techniques and can be easily incorporated in a plug-and-play manner, we choose three baselines [26, 36, 30] to combine with our framework for evaluation: PGD-AT-ANCRA, TRADES-ANCRA, and MART-ANCRA.

## 4.1   Settings

**Implementation**   On CIFAR-10 and CIFAR-100 [22], we train ResNet18 [15] with a weight decay of $2.0 \times 10^{-4}$. On Tiny-ImageNet [10], we use PreActResNet18 [16]with a weight decay of $5.0 \times 10^{-4}$. We adopt the SGD optimizer with a learning rate of 0.01, a momentum of 0.9, epochs of 120 and a batch size of 128 as [30]. For the trade-off hyperparameters $\beta$, we use 6.0 in TRADES and 5.0 in MART, following the original setting in their papers. For other hyperparameters, we tune the values based on TRADES-ANCRA. We generate adversarial example for training by $L_{\infty}$-norm PGD [26], with a step size of 0.007, an attack iterations of 10 and perturbation budget of 8/255. We use single NVIDIA A100 and two GTX 2080 Ti in the experiments.

**Baseline**   We compare the proposed PGD-AT-ANCRA, TRADES-ANCRA, and MART-ANCRA with the popular baselines: PGD-AT [26], TRADES [36], MART [30] and SAT [18]. Moreover, we also choose three state-of-the-art methods: AWP [31], S2O [20] and UDR [4]. We keep the same settings among all the baselines with our settings and follow their original hyperparameters.

**Evaluation**   We choose several adversarial attacks to attack the target models, including PGD [26], FGSM [13], C&W [5] and AutoAttack [9] which is a powerful and reliable attack and an ensemble attack with three white-box attacks and one black-box attack. We notice that our methods use the auxiliary probability vector $p$ in the training and testing phase, so we design two scenaios: 1) train with $p$ and test without $p$; 2) train with $p$ and test with $p$. 1) denotes evaluation against white-box attacks and 2) denotes evaluation against adaptive attacks. Following the default setting of AT, the max perturbation strength is set as 8. / 255. for all attack methods under the $L_{\infty}$. The attack iterations of PGD and C&W is 40 (i.e., PGD-40), and the step size of FGSM is 8. / 255. unlike 0.007 for other attacks. The clean accuracy and robust accuracy are used as the evaluation metrics.

## 4.2   Comparison results of feature distribution

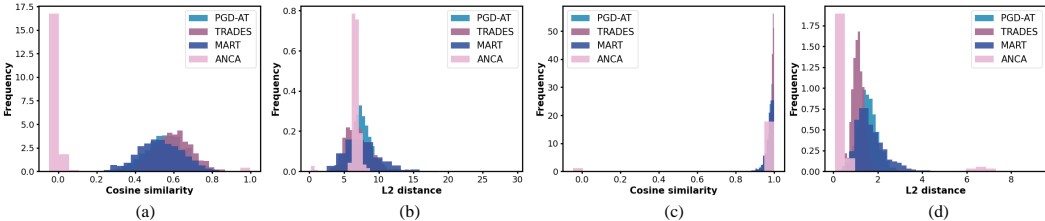

Figure 3: Frequency histograms of the $L_2$ distance and cosine similarity of feature of natural examples, AEs and OEs. We train ResNet-18 models on CIFAR-10 with four defense techniques: PDG-AT, TRADES, MART and TRADES-ANCRA. Other details are the same with Figure 1

Frequency histograms of feature distribution is shown in Figure 3. It is shown that our methods can greatly improve feature distribution, which follows the characteristics of *exclusion* and *alignment*. In Figure 3 (a) and Figure 3 (b), it shows that the cosine similarity of the model trained by our method between natural examples and OEs shows a skewed distribution between -0.05 and 0.1, and the $L_2$

distance with our method shows a Gaussian distribution between 5.5 and 10.0, which indicates natural examples and OEs have been fully distinguished in the feature space and *exclusion* has been met. In Figure 3 (c) and Figure 3 (d), it shows that in the model trained by our method there are a uniform distribution between 0.95 and 0.99 for the cosine similarity of the feature between natural examples and AEs, and a skewed distribution between 0.05 and 1.5 for the $L_2$ distance of the feature, which indicates the feature between natural examples and AEs is very close to each other and *alignment* has been confirmed. Thus, our framework successfully helps AT to obtain robust feature.

## 4.3 Comparison results against white-box attacks

Table 1: Robustness (%) against white-box attacks. Nat denotes clean accuracy. PGD denotes robust accuracy against PGD-40. FGSM denotes robust accuracy against FGSM. C&W denotes robust accuracy against C&W. AA denotes robust accuracy against AutoAttack. Mean denotes average robust accuracy against these four attacks. We show the most successful defense with **bold**.

| Defense | CIFAR-10 | | | | | | CIFAR-100 | | | | | |
|---|---|---|---|---|---|---|---|---|---|---|---|---|
| | Nat | PGD | FGSM | C&W | AA | Mean | Nat | PGD | FGSM | C&W | AA | Mean |
| PGD-AT | 80.90 | 44.35 | 58.41 | 46.72 | 42.14 | 47.91 | 56.21 | 19.41 | 30.00 | 41.76 | 17.76 | 27.23 |
| TRADES | 78.92 | 48.40 | 59.60 | 47.59 | 45.44 | 50.26 | 53.46 | 25.37 | 32.97 | 43.59 | 21.35 | 30.82 |
| MART | 79.03 | 48.90 | 60.86 | 45.92 | 43.88 | 49.89 | 53.26 | 25.06 | 33.35 | 38.07 | 21.04 | 29.38 |
| SAT | 63.28 | 43.57 | 50.13 | 47.47 | 39.72 | 45.22 | 42.55 | 23.30 | 28.36 | 41.03 | 18.73 | 27.86 |
| AWP | 76.38 | 48.88 | 57.47 | 48.22 | 44.65 | 49.81 | 54.53 | 27.35 | 34.47 | 44.91 | 21.98 | 31.18 |
| S2O | 40.09 | 24.05 | 29.76 | 47.00 | 44.00 | 36.20 | 26.66 | 13.11 | 16.83 | 43.00 | 21.00 | 23.49 |
| UDR | 57.80 | 39.79 | 45.02 | 46.92 | 34.73 | 41.62 | 33.63 | 20.61 | 24.19 | 33.77 | 16.41 | 23.75 |
| PGD-AT-ANCRA | **85.10** | **89.03** | 87.00 | **89.23** | 59.15 | 81.10 | 59.73 | 58.10 | 58.45 | 58.58 | 34.44 | 52.39 |
| TRADES-ANCRA | 81.70 | **82.96** | 82.74 | 83.01 | **59.70** | 77.10 | 53.73 | 51.24 | 52.17 | 52.55 | **35.81** | 47.94 |
| MART-ANCRA | 84.88 | 88.56 | **87.95** | 88.77 | 59.62 | **81.23** | **60.10** | **58.40** | **58.74** | **59.41** | 35.05 | **52.90** |

We train ResNet-18 by different defense on CIFAR-10 and CIFAR-100 to evaluate them under white-box attacks. And more results in PreActResNet18 on Tiny-ImageNet are provided in our supplementary material. The results on CIFAR-10 and CIFAR-100 are shown in Table 1. First, on CIFAR-10, our approaches improve the clean accuracy of based approaches by 5.2%, 3.2% and 5.9%, and also improves the robust performance under all the attacks (e.g., increase by 44.7%, 34.6% and 39.7% against PGD). Compared with state-of-the-art defense, the robust accuracy against different attacks of our methods is almost two times as large than theirs (e.g., 81.23% VS 49.81%). Second, on CIFAR-100, our approaches also greatly improve the robustness and advance the clean accuracy. The clean accuracy of our methods has been increased by 3.5%, 0.3% and 6.8% compared with based methods, and the lowest average robust accuracy of ours is larger than the best one among other methods by 16.8%. In general, our three approaches gain the best performance both in the natural and attacked scenaios. To our surprise, MART-ANCRA and PGD-ANCRA rather than TRADES-ANCRA gain the best performance in a lot of cases without hyper-parameter tuning. Besides, our approaches not only improves robustness but also enhances clean accuracy, though there is always a trade-off between clean and robust accuracy. These results indicate that our approaches can vastly boost the robustness of models against white-box attacks.

## 4.4 Comparison results against adaptive attacks

We train several ResNet18 models on CIFAR-10 by PGD-AT-ANCRA, TRADES-ANCRA, MART-ANCRA and test the same models without $p$. In addition, we report vanilla based approaches as baseline. Results are in Table 2. It indicates that our approaches can still maintain superb performance after adaptive attacks, e.g., the robust accuracy against PGD of our methods without $p$ are larger than those of baseline by 13.28%, 10.08% and 8.06%.

## 4.5 Ablation studies

**Two defense methods.** We train four models by TRADES, TRADES with the asymmetric negative contrast (TRADES-ANC), TRADES with the reverse attention (TRADES-RA) and TRADES-ANCRA, respectively. The results of evaluation against adaptive attacks are shown in Table 3. First,

Table 2: Robustness(%) of ResNet-18 trained with our approaches and attacked with or without $p$.

| Approach | Nat | Attack with $p$ | | | Attack without $p$ | | |
|---|---|---|---|---|---|---|---|
| | | PGD | FGSM | C&W | PGD | FGSM | C&W |
| Vanilla TRADES | 78.92 | \ | \ | \ | 48.40 | 59.60 | 47.59 |
| TRADES-ANCRA | 81.70 | 61.68 | 61.56 | 72.36 | 82.96 | 82.74 | 83.01 |
| Vanilla PGD-AT | 80.90 | \ | \ | \ | 44.35 | 58.41 | 46.72 |
| PGD-AT-ANCRA | 85.10 | 54.43 | 58.23 | 66.36 | 89.03 | 87.00 | 89.23 |
| Vanilla MART | 79.09 | \ | \ | \ | 48.90 | 60.86 | 45.92 |
| MART-ANCRA | 84.88 | 56.96 | 60.43 | 71.06 | 88.56 | 87.95 | 88.77 |

when incorporating the asymmetric negative contrast only, the performance of robustness against all the attacks and clean accuracy have been improved compared with vanilla TRADES (e.g., 48.36% VS 54.18% against PGD-40). Next, when incorporating the reverse attention only, the performance on clean and adversarial data is also improved greatly compared with TRADES (e.g., 48.36% VS 61.69% against PGD-40). Thus, it shows each method contributes to robustness and generalization. Besides, when Trdeas-ANCRA is compared with TRADES-RA, the clean accuracy and robust accuracy against all the attacks except AA have been enhanced, which indicates that the two strategies are compatible and the combination can alleviate the side effect of independent methods.

**Strategy of negative samples**   We compare our strategy of the targeted attack with other strategies to select negative samples, including Random, Soft-LS and Hard-LS proposed by Bui et al. [3]. The details of them are provided in our supplementary material. The results are shown in Table 4. To make a comprehensive compare, we show results of both the best models and last models with different strategies. It shows that our strategy have the best performance of robustness and clean accuracy in the last models, and achieve the best robust accuracy in the best models.

Table 3: Clean and robust accuracy (%) of ResNet-18 trained by TRADES, TRADES-ANC, TRADES-RA and TRADES-ANCRA on CIFAR-10 against various attacks.

| Defense | Nat | PGD | FGSM | C&W | AA |
|---|---|---|---|---|---|
| TRADES | 78.92 | 48.40 | 59.60 | 47.59 | 45.44 |
| TRADES-ANC | 80.77 | 54.18 | 63.44 | 49.84 | 48.51 |
| TRADES-RA | 80.46 | 61.59 | 61.48 | 72.15 | 61.02 |
| TRADES-ANCRA | 81.70 | 61.68 | 61.56 | 72.36 | 59.70 |

Table 4: Results of the best and last with four strategies of negative example. Best- denotes results in the best models and Last- denotes results in the last models. We show the best results with **bold**.

| Strategy | Best-Nat | Best-PGD | Last-Nat | Last-PGD |
|---|---|---|---|---|
| Random | 81.44 | 62.64 | 81.78 | 61.71 |
| Soft-LS | 82.10 | 61.83 | 80.62 | 58.47 |
| Hard-LS | **82.30** | 62.53 | 82.13 | 60.98 |
| Targeted attack | 81.36 | **63.08** | **82.18** | **62.02** |

# 5   Conclusion

This work addresses an overlook of robust representation learning in the adversarial training by a generic AT framework with the asymmetric negative contrast and reverse attention. We propose two characteristics of robust feature to guide the improvement of AT, i.e., *exclusion* and *alignment*. Specifically, the asymmetric negative contrast based on probabilities fixes natural examples, and only pushes away adversarial examples of other classes in the feature space. Besides, the reverse attention weights feature by parameters of the linear classifier, to provide class information and align feature of the same class. Our framework can be used in a plug-and-play manner with other defense methods. Analysis and empirical evaluations demonstrate that our framework can obtain robust feature and greatly improve robustness and generalization.

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
