# 1 Supplementary Material

## 1.1 Experiments on Tiny-ImageNet

We train PreActResNet18 [3] models on Tiny-ImageNet [1]. We adopt the SGD optimizer with a learning rate of 0.1, a momentum of 0.9, a weight decay of $5.0 \times 10^{-4}$, epochs of 120 and a batch size of 128. We choose the same hyperparameters as our settings in the text. And we also generate adversarial examples for training by $L_\infty$-norm PGD [5], with a step size of 0.007, an attack iterations of 10 and perturbation budget of 8/255.

Table 1: Clean and robust accuracy (%) of PreActResNet18 trained by PGD-AT-ANCRA, TRADES-ANCRA and MART-ANCRA on Tiny-ImageNet. PGD denotes robust accuracy against PGD-40, Adaptive PGD denotes robust accuracy against adaptive PGD-40.

| Defense | Nat | PGD | Adaptive PGD |
|---|---|---|---|
| PGD-AT-ANCRA | 35.18 | 26.73 | 14.99 |
| TRADES-ANCRA | 33.47 | 25.83 | 13.54 |
| MART-ANCRA | 34.43 | 26.65 | 15.03 |

## 1.2 Experiments of feature visualization

We conduct several experiments of feature visualization on CIFAR-10 [4], in the ResNet18 [2] models trained by PGD-AT [5], TRADES [8], MART [7] and Trades-ANCRA. We use UMAP [6] to reduce the dimension of feature vectors and draw the distribution map. Results are shown in Figure 2 and Figure 1, where different colors denotes samples of different classes. Unlike traditional AT methods, our approach can improve feature distribution by pulling close samples of the same class and pushing away samples of different classes, which follows *exclusion* and *alignment*.

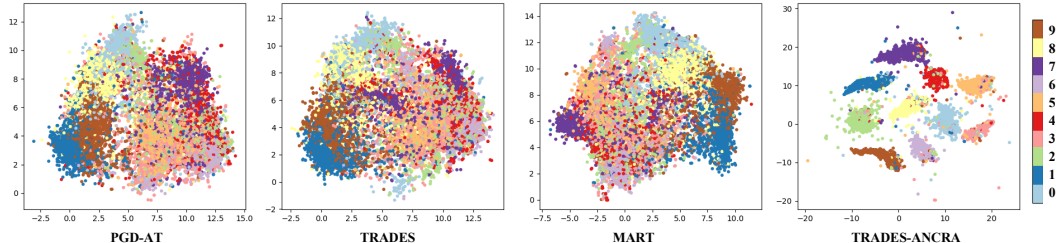

Figure 1: Feature visualization of four methods on natural and adversarial examples. Adversarial samples are crafted by PGD-10.

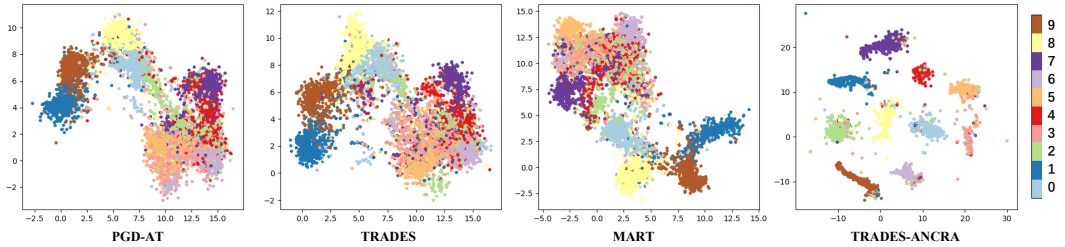

Figure 2: Feature visualization of four methods on natural examples.

## 1.3 Experiments about hyperparameters

We have used two hyperparameters in the loss function: $\alpha$ and $\zeta$. $\alpha$ denotes the weighting factor to adjust the magnitude of the two repulsive forces, which we mentioned in Section 3.2.1 in the

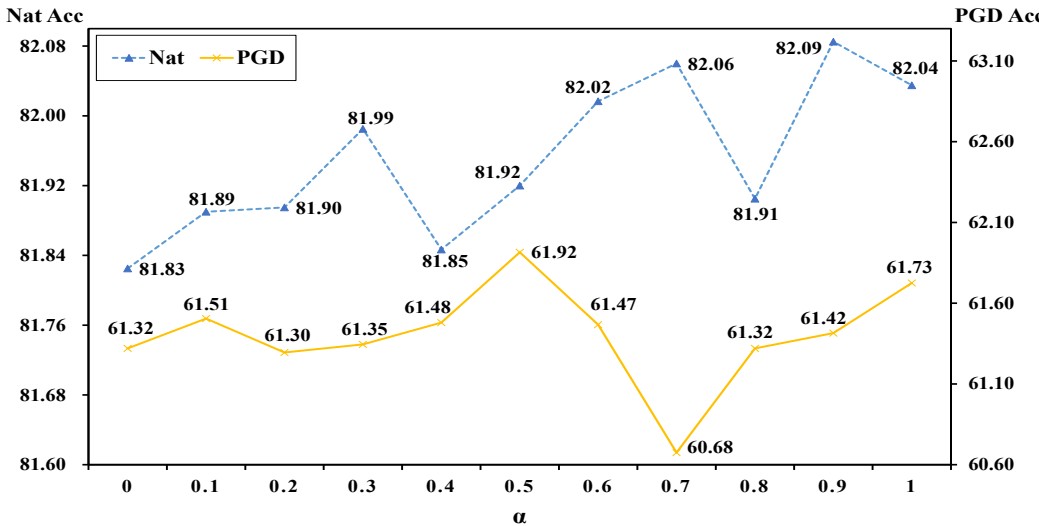

Figure 3: Clean and robust accuracy with different $\alpha$.

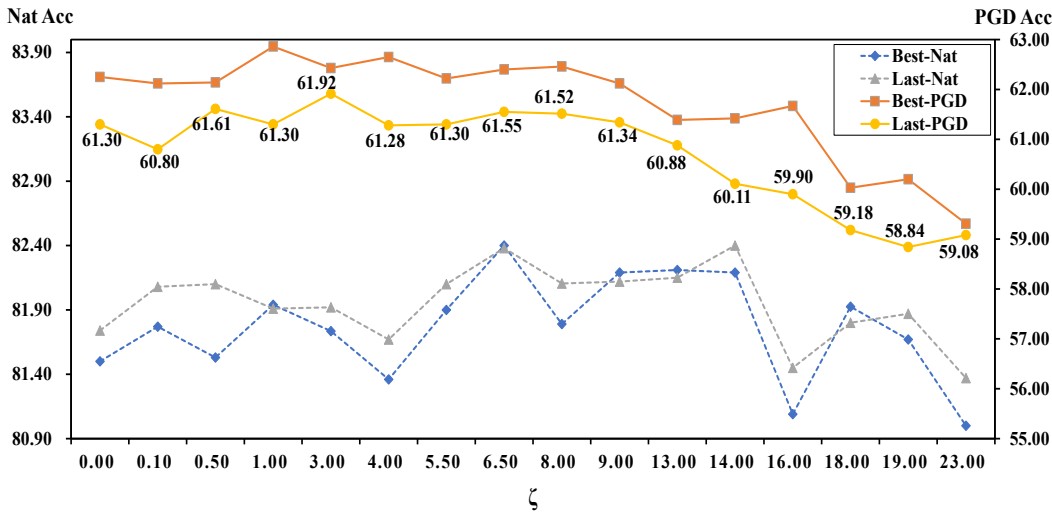

Figure 4: Clean and robust accuracy with different $\zeta$.

text. And $\zeta$ denotes the weight of the asymmetric negative contrast in the whole loss. We tune these hyperparameters on CIFAR-10 in the ResNet18 models as follows:

In Figure 3, it shows that there is a positive relationship between the accuracy and $\alpha$. Though there is an obvious trade-off between the clean accuracy and robust accuracy when $\alpha$ equals from 0.5 to 1.0, we can still see an abnormal increasing trend. It is because the larger $\alpha$ leads to the larger repulsive force from the OE to the natural example, to prevent the natural example from being pushed into the wrong class. Besides, in Figure 4, we choose $\zeta = 3.00$ in which models gain the best robust accuracy against PGD-40 in the last epoch.

## 1.4 Details of strategies of negative samples

We have totally tested four strategies of negative samples as follows. During training, all of them get samples from current batch. The first three methods choose appropriate negative samples by labels or predicted classes, while our approach attacks samples of labels different from natural examples', and gain adversarial examples predicted as classes of natural examples.

Table 2: Details of strategies of negative samples. $h(\cdot)$ denotes predicted class, $gt(\cdot)$ denotes the label of input, $ne$ denotes natural examples, OE denotes the example of the other class and $\mathbb{N}(x, \epsilon)$ denotes the neighborhood of $x : \{\tilde{x} : \|\tilde{x} - x\| \leq \epsilon\}$, respectively.

| Strategy | Condation |
|---|---|
| Random | $\{OE\|gt(OE) \neq gt(ne)\}$ |
| Soft-LS | $\{OE\|gt(OE) \neq gt(ne), h(OE) = h(ne)\}$ |
| Hard-LS | $\{OE\|gt(OE) \neq gt(ne), h(OE) = gt(ne)\}$ |
| Targeted attack | $\{OE'\|gt(OE) \neq gt(ne), OE' = \underset{\mathbb{N}(OE,\epsilon)}{max} \left( \mathcal{L}_{CE} \left( (f(OE), gt(ne)) \right) \right)\}$ |