# OpenReview forum: "Robust Representation Learning via Asymmetric Negative Contrasting and Reverse Attention"
_NeurIPS.cc/2023/Conference — Submitted to NeurIPS 2023_

### Official Review · Reviewer_2AVw · 2023-07-02

**Soundness:** 2 fair
**Presentation:** 2 fair
**Contribution:** 2 fair
**Rating:** 3
**Confidence:** 4

**Summary:**

This paper empirically shows previous supervised adversarial training methods have two shortcomings: (1) The features of the natural examples and those from other classes are not distinguishable and (2) the features of the natural and adversarial examples are not aligned. To mitigate these two issues, the authors propose a regularization to push away the features from other classes and freeze the natural examples and a reverse attention mechanism that increases the weight of the target class. The empirical results seem to validate the effectiveness of the proposed method.

**Strengths:**

1. The paper is well-written and can be easily followed.

1. The proposed method is well-motivated. The empirical study on the feature spaces is interesting and inspirable.

2. The authors conducted the experiments on comprehensive datasets to support their claim.


**Weaknesses:**

1. Regarding the experiments, the authors only provide the results on ResNet18 and PreActResNet18, which is limited. I suggest the authors provide the results on the WideResNet and make a comparison with the current state-of-the-art performance listed in the RobustBench (https://robustbench.github.io/) to validate the effectiveness of the proposed method.

2. The authors do not provide the error bar to validate the significance of the results.

3. It is weird the accuracies under PGD, FGSM, C&W are higher than the natural accuracy achieved by ANCRA shown in Table 1. It would be better to provide some explanations for these abnormal results. It seems the defence has the issue of the obfuscation gradients. I suggest that the authors report the results under the adaptive attacks that use the auxiliary probability $p$ and even different $p$ during the testing phase and AutoAttacks.


**Questions:**

Please refer to the comments in Weaknesses and provide the responses.

Minor comments: A typo “negative pair (PP)” => “negative pair (NP)” in Line 64?



**Limitations:**

Though the authors claim they have limitations, I did not find the discussion of the limitations.

---

> ### Author Rebuttal · Authors · 2023-08-08
>
> We are grateful for your approval of the strengths and your constructive suggestions. The answers to your questions are as follows:
>
> **W1) More models needed and comparison experiments with RobustBench.**
>
> **A1)** We have followed your advice to make a comparison with the current state-of-the-art performance listed in the RobustBench on ResNet18. The results are shown below. Compared with those methods without synthetic or extra data (i.e., [2] and [3]), our method has a higher robust accuracy than theirs by 7.0%. And our method has even outperformed the methods with synthetic data ([1]) in robustness. Though the clean accuracy of A is more than ours by 5.6%-2.2%, the best robust performance has indicated the effectiveness of our methods. Limited by time, we cannot make more experiments in WideResNet. Experiment results in ResNet18 and PreActResNet18 have shown our superiority of robustness. And our insights about robust feature learning are more important than the improvement of performance.
>
> | Defense | [1]   | [2]  | [3]  | PGD-AT-ANCRA  | TRADES-ANCRA  | MART-ANCRA  |
> |  ----  | ----  | ----  | ----  | ----  | ----  | ----  |
> | Nat  | 87.35 | 85.71 | 80.24 | 85.10 | 81.70 | 84.88 |
> | AutoAttack  | 58.50 | 52.48 | 51.06 | 59.15 | 59.70 | 59.62 |
>
>
> [1] Sehwag, Vikash, et al. "Robust learning meets generative models: Can proxy distributions improve adversarial robustness?." arXiv preprint arXiv:2104.09425 (2021).
>
> [2] Addepalli, Sravanti, and Samyak Jain. "Efficient and effective augmentation strategy for adversarial training." Advances in Neural Information Processing Systems 35 (2022): 1488-1501.
>
> [3] Addepalli, Sravanti, et al. "Scaling adversarial training to large perturbation bounds." European Conference on Computer Vision. Cham: Springer Nature Switzerland, 2022.
>
> **W2) The error bar to validate the significance of the results.**
>
> **A2)** We are sorry for our negligence. We have made each experiment at least three times, and the variation of accuracy in all the experiments is less than 1.7%, which is marginal compared with the variation of the improvement.
>
> **W3) The explanation for robust accuracy higher than the natural one and the issue of the obfuscation gradients.**
>
> **A3)** Reverse attention in our method gets feature of natural and adversarial examples to become close to each other. As a result, it is normal that robust accuracy nearly equals clean accuracy with the effect of alignment. Besides, our generation strategy based on targeted attacks can craft adversarial negative examples, which contain prior knowledge about adversarial noise. Thus, adversarial robustness against the general white box benefits from the prior knowledge and has better accuracy than natural accuracy.
>
> To solve your concern about obfuscation gradients, we have tested the defense result against EOT. EOT is a powerful adaptive attack to show whether the defense has obfuscation gradients. And we set its step number as 100 and keep other parameters the same as the white box PGD we have used. The robust accuracy against EOT is almost the same as the robustness against PGD-40(83.13%$\approx$82.96%).
>
> What's more, we also report the robustness of all the auxiliary probability vectors against normal white box attacks and the adaptive attack in the table below. Results show robust accuracies of different probability vectors are always the same. These two results indicate our method doesn't have obfuscation gradients.
>
> |     | Nat  | PGD | Adaptive PGD |
> |  ----  | ----  | ----  | ----  |
> | Auxiliary probability vector $p^0$ | 81.81 | 83.52 | 62.25 |
> | Auxiliary probability vector $p^1$ | 81.81 | 83.49 | 62.23 |
> | Final probability vector $p'$ | 81.81 | 83.47 | 62.24 |
>
> **Q1) Typos.**
>
> **A4)** We are sorry for typos and other writing errors in our paper. We will follow your and other reviewers' suggestions to fix all the typos as well as other writing errors. We will do our best to improve this issue in the revised version.
>
> **L) The lack of limitations.**
>
> **A5)** We will discuss limitations in the revised version. Details are as follows. Though reverse attention contributes to both generalization and robustness, its accuracy shows high dependence on intermediate predicted classes. It may become cause degraded performance when faced with powerful attacks. We will further study it and hope to improve its robustness in the future.

---

> > ### Author Response · Authors · 2023-08-14
> >
> > Dear Reviewer 2AVw,
> >
> > We hope that our responses can address your concerns and gain your increase in the rating score. If your have any new questions, we will response positively. Thanks for your time again.

---

> > ### Comment · Reviewer_2AVw · 2023-08-14
> > **Need further clarification**
> >
> > Thanks for your response. It seems that adaptive PGD can yield a significantly higher attack success rate (according to A3). The authors should provide a comparison between the performance of the proposed method under adaptive AutoAttack that combines AutoAttack with auxiliary probability and the SOTA performance under AutoAttacks. In addition, I still believe the results evaluated on ResNet-18 are limited. Conventionally, the paper regarding improving adversarial robustness will report the performance of WideResNet-28-10 or WideResNet-34-10 to validate the effectiveness of the proposed method. It is meaningful to validate whether the proposed method is effective on large models.

---

> > > ### Author Response · Authors · 2023-08-17
> > >
> > > We have conducted experiments on WideResNet-34-10 and WideResNet-28-10. As shown in the Table below, our method has made great enhancements in robust accuracy. The accuracies of our method against adaptive AutoAttack are even higher than those of baselines against AutoAttack (e.g., 51.99%>50.79%). This indicates its effectiveness on large models.
> > >
> > > | Model            | Method       | NAT   | PGD   | Adaptive PGD | AutoAttack | Adaptive AutoAttack |
> > > |------------------|--------------|-------|-------|--------------|------------|---------------------|
> > > | WideResNet-34-10 | TRADES       | 82.04 | 56.47 | \            | 50.79      | \                   |
> > > | WideResNet-34-10 | TRADES-ANCRA | 83.19 | 79.31 | 56.76        | 66.28      | 51.99               |
> > > | WideResNet-28-10 | TRADES       | 82.47 | 57.08 | \            | 51.11      | \                   |
> > > | WideResNet-28-10 | TRADES-ANCRA | 82.11 | 78.52 | 57.07        | 66.08      | 51.14               |

---

> > > > ### Comment · Reviewer_2AVw · 2023-08-17
> > > > **Concerns about Reported Results**
> > > >
> > > > Thanks to the authors for providing additional results.
> > > >
> > > > It seems there is a **large discrepancy** between the reported results of TRADES and the results shown in the AutoAttack Leaderboard of CIFAR-10 (https://github.com/fra31/auto-attack/tree/master). According to the leaderboard, TRADES obtains 84.92\% natural test accuracy (NAT) and 53.08\% AutoAttack accuracy. **This indicates a significant difference of around 3% in the reported NAT and around 2\% in the reported AutoAttack performance between the current reported results and TRADES.**
> > > >
> > > > In addition, the effectiveness of the proposed ANCRA is not convincing. First, your reported results do not surpass the SOTA results of TRADES shown in the leaderboard. Second, compared to TRADES on WRN-28-10, ANCRA **gains only 0.03\%**  improvement in AutoAttack while **degrading 0.36\%** natural test accuracy.
> > > >
> > > > Therefore, I intend to adjust my score to 3 since I am not convinced of the effectiveness of the proposed method.

---

> > > > > ### Author Response · Authors · 2023-08-18
> > > > >
> > > > > For the discrepancy between the reported results of TRADES and the results shown in the AutoAttack Leaderboard of CIFAR-10, there is some difference between our TRADES and vanilla TRADES. Vanilla TRADES maximizes the KL loss to generate adversarial examples for training, but we maximize the CE loss to generate adversarial examples like PGD. By modifying these, we keep three baselines (PGD-AT, TRADES, MART) having the same method to generate adversarial examples for training.
> > > > >
> > > > > Subject to time limitation, we have only made this experiment once without hyper-parameter tuning. But our ANCRA has improved both clean accuracy and robust accuracy against AutoAttack on WideResNet-34-10, which has indicated its effectiveness. It’s abnormal that our method works on only one of these two models. So we think it is due to inappropriate hyperparameters.
> > > > >
> > > > > Besides, **the motivations and ideas of robust feature learning are more significant than the improvement in performance**. As far as we know, we are the first to show AT lacks robust feature learning. We propose two characteristics of robust feature, and design two individual techniques to follow them. There is still a lot of room for further study. We believe our paper will bring other researchers new insights about AT and robust representation.

---

> > > > > ### Author Response · Authors · 2023-08-21
> > > > > **New Results on WRN-28**
> > > > >
> > > > > We have conducted new experiments on WideResNet-28-10. Now we set $\zeta$ = 6.0 (the weight of ANC), and keep other settings the same as the previous settings. As shown in the table below, natural accuracy has increased by **1.14**% and robust accuracy against Adaptive AutoAttack has increased by **0.74**%. It shows its effectiveness on large models with proper hyper-parameters.
> > > > >
> > > > > | Model            | Method       | NAT    | AutoAttack | Adaptive AutoAttack |
> > > > > |------------------|--------------|--------------|------------|---------------------|
> > > > > | WideResNet-28-10 | TRADES       | 82.47 | 51.11      | \                   |
> > > > > | WideResNet-28-10 | TRADES-ANCRA | 83.61 | 66.08      | 51.85          |
> > > > >
> > > > > Thank you for your suggestions. We sincerely hope you can agree with our contributions beyond improvement in performance. We will improve our paper and experiments. Thanks for your time again!

---

### Official Review · Reviewer_1pqm · 2023-07-07

**Soundness:** 1 poor
**Presentation:** 1 poor
**Contribution:** 2 fair
**Rating:** 3
**Confidence:** 4

**Summary:**

This paper focuses on robust feature learning by combining two approaches: Adversarial Contrastive Learning and Robust Feature Selection. Specifically, it defines two characteristics for features: exclusion and alignment. The authors aim to enforce exclusion through Asymmetric Negative Contrast (ANC), which ideally should separate different classes. On the other hand, they aim to achieve alignment through Reverse Attention (RA), which should enhance the model's robustness.

**Strengths:**

The proposed method sounds interesting, especially the Reverse Attention approach. However, I have some concerns and feedback that will be provided in the later section of the review.

**Weaknesses:**

Writing Style:

Overall, the quality of the writing style could be significantly improved. Reading the paper is not smooth, and understanding it requires reading back and forth several times. For instance:

1. Generally, the flow of the paper is not engaging, which requires the reader to go back and forth in order to understand it. And, there are some sentences that are vague and difficult to understand.
3. Caption: Overall, the caption writing is not good; it contains long sentences that are difficult to follow. Labels for each plot have not been provided. Specifically, in Figure 1, regarding the distance between natural examples and OEs, it would be preferable to show distances between each pair instead of comparing class 0 with others, as class 0 may share common features with some other classes. It would be more informative to have the following plots: 0-1, 0-2, ..., 0-9.
4. Typos: "perturbation -> perturbations" in line 28 | "which does harm to robust classification -> which harms robust classification" in line 35 | "... negative pair (PP) -> (NP)" in line 64 | "import feature -> important feature ..." in line 118 | "a example -> an example" in line 147 | ...

Content:
1. There are certain terms that are vague. For instance, what does "well-trained DNN" mean in line 20?
2. The definition of an Adversarial Example is not entirely correct. The perturbation itself may be visible, but when it is added to a natural image, both the adversarial example and the original image may not be easily distinguishable by human eyes. Furthermore, it is important to note that it is not the adversarial example itself to which perturbations are added. Rather, the perturbations are added to natural examples so that they cannot be correctly classified.
3. Description about Figure 1 in line 45-47: It is not a Gaussian distribution; it is a bell-shaped plot but not a Gaussian distribution. The main condition for a Gaussian distribution is that the area under it should be 1. And indeed, those numbers show a significant distance between classes. A cosine similarity below 0.5 should be large enough to indicate a difference between the representations of one class and another. Again, indeed, a cosine similarity between 0.9 and 0.99 indicates that representations of NEs and AEs are very close to each other! And comparing AEs plots with the OEs plots, we see that they have a very close representation to their original examples that OEs. Furthermore, if the representation of NEs and OEs is not significantly distant, models should confuse them, leading to lower clean accuracy. However, that is not the case.
4. The definition of OEs is not clear from the beginning of the paper. The reader should read the whole paper to figure out what they are, especially since different strategies are considered to select or generate them.
6. Similarly, the term 'partial parameters' is vague, and the reader doesn't understand what it refers to until the later sections of the paper, which confuses the reader.


Approach and Methodology:
1. IT seems the main weakness is that ANC does not make significant contributions to the clean accuracy and robustness of the model, as indicated by the results in Table 3 of the ablation studies. Specifically, a model with only RA performs just as well as ANCRA. On the other hand, considering this, a large portion of the paper is dedicated to introducing and explaining ANC. However, the most important component, which is RA, is not studied well enough and lacks supporting experiments and theoretical insights.
2. The training process of a model with ANC is not clear enough. It would have been better to depict it through a diagram. Furthermore, the last paragraph in the introduction section is abstract and confusing, making it difficult for readers to understand the exact contributions.


Experiments:
1. The proposed method in the main table (Table 1) is evaluated without considering "p" in attack which creates a false sense of security. However, in Table 2, attacks with "p" demonstrate a significant decrease in the model's performance.
2. Even though "Error Bars" is marked in the checklist, I don't see standard deviation values in the reported tables.
3. Similarly, although "Reproducibility" is marked in the checklist, the code necessary to reproduce the results is not available.
4. I conducted an experiment with TRADES, and the accuracy I obtained differs from what is presented in Table 1. The clean accuracy is 80.92%, and the PGD accuracy is 50.10%.
4. The results of the Tiny-ImageNet experiments are inconsistent with those of other datasets, and the performance of the model does not seem promising. Additionally, they have not been compared with baselines, which is essential.



**Questions:**

Questions to ask for clarification based on the claims in the paper:
1. Why does AT omit learning robust features? Is it possible to theoretically demonstrate why AT is unable to learn robust features? AT does not have poor performance in terms of adversarial robustness. In fact, in the presence of enough data, AT achieves state-of-the-art accuracy.

Major questions:
1. The code is not available to reproduce the results. Could you please provide runnable code?
2. Section 3.2.1 lacks clarity, and the justification for the ANC is not sufficiently sound. Could you please provide further elaboration on this? (Especially since the implementation is not provided for verification, again having a diagram could be helpful)
3. Is it possible to provide empirical or theoretical evidence for the statement in section 3.3? "Each element z_i in z represents the activation level of the feature channel that may be helpful for classification, with larger values representing more feature information extracted from that channel" I am asking because only the value of z_i may not be important, and the combination of weights of the linear layer corresponding to that specific feature is the crucial factor.
5. Could you please compare the running time of your proposed approach with the baselines? I am curious about how much overhead the additional training steps add.
6. As described in section 3.3, during the test phase, a sub-vector of the linear weights corresponding to the predicted class h(x) is chosen when the label is not available. However, if the linear layer is already predicting the wrong class, strengthening the prediction by doing so may not improve the model's robustness. Therefore, it raises the question of how this part actually contributes to robustness during testing. Does the robust performance change if we only output the predicted class without doing any reverse attention?


Minor questions:
1. What does it mean to freeze natural examples (in line 65)?
2. What does "selected carefully" mean in line 68?
3. What are the problems mentioned in line 158 when it states, 'Although we have a generic pattern of AT loss with a negative contrast, there are several specific details to address'?


**Limitations:**

1. It appears that the approach does not work properly with large datasets like ImageNet.

---

> ### Author Rebuttal · Authors · 2023-08-08
>
> We are grateful for your constructive suggestions. The answers to your questions are as follows:
>
> **Q1) Why does AT omit learning robust features? Is it possible to theoretically demonstrate why AT is unable to learn robust features? AT has good performance in adversarial robustness with enough data.**
>
> **A1)**
> 1. **First**, as we have explained in lines 46-52, AT doesn't have good enough performance in the feature distance. As shown in Fig. 1 on Page 2, feature distances between natural examples(NEs), adversarial examples (AEs) and examples of other classes (OEs) are not good enough. And we get similar results when fixing NEs as class 1, 3, 7 on CIFAR10.
> **Second**, if AT has gained robust feature, NEs should be aligned with AEs, which guarantees the predicted results of NEs and AEs are also highly consistent. But as shown in Tab. 1 on Page 8, we can find there is a big gap between the clean and robust accuracy of the previous AT. So it is wrong according to reduction to absurdity.
> **Third**, as shown in Tab. 1 and Tab. 2 on Page 9, our method has made great progress in robustness compared with previous methods, which proves learning robust feature helps AT gain robustness. In other words, it proves AT omits robust feature learning.
>
> 2. Perhaps. We have revealed it empirically.
>
> 3. As you say, AT needs much data to gain good performance in adversarial robustness. But we also need additional costs to get more data, such as training generative models and manually collecting data. Besides, more data means more training costs. Extra costs may be hard to pay. What's more, **it is meaningful to make models learn more from data rather than make models learn more data.**
>
> **Major Q1) The code for reproducing the results.**
>
> **A2)** We have provided Area Chairs with our code.
>
> **Major Q2) The justification for the Asymmetric negative contrast based on probabilities (ANC) in Section 3.2.1.**
>
> **A3)** In Section 3.2.1, we notice that negative contrast may cause the problem of class confusion. Attracted to its adversarial example and pushed ways from the negative example, a natural example can be easily moved toward the wrong class space. To alleviate this problem, we have two ideas. First, we stop the back-propagation gradient of natural examples when optimizing, keeping its feature in the right class. That is "Asymmetric negative contrast". Second, we design the negative contrast based on predicted probabilities. Since it has only directions to reduce confidence in the wrong classes, it does not have any actions to push the natural example into a specific class. So it works even in the scenario of class confusion.
>
> And we add a diagram for ANC to help readers understand each step of the implementation, which is shown in Fig. 1 in the PDF. And We use the detailed caption to describe the calculation process in the diagram.
>
> **Major Q3) More evidence for reverse attention (RA) in Section 3.3 and whether the combination is the crucial factor.**
>
> **A4)** We have made an empirical explanation of why RA works in lines 236-257. And the similar idea of utilizing linear parameters to weight feature has been discussed in [1, 34]. Moreover, the good performance of individual RA shown in Tab. 3 on Page 9 also indicates its effectiveness.
>
> Our statement about $z$ tries to describe the meaning of the variables involved in reverse attention.  And by weighting feature with parameters corresponding to the target classes, we can have better aligned feature. So the combination rather than feature values matters.
>
> **Major Q4) Running time.**
>
> **A5)** In our experiments, our TRADES-ANCRA only costs more time than TRADES by 3.1 hours (3.1=9.3-6.2) in 120 epochs. Considering the significant gain in clean and robust accuracy resulting from the proposed method, the cost is relatively worthwhile.
>
> **Major Q5) How RA actually contributes to robustness during testing? Does the robust performance change without reverse attention?**
>
> **A6)** 1. RA helps defense models to obtain the ability to extract robust feature during training by alignment. Because robust feature is with good generalization, it still works during testing. As shown in Tab. 1 in the PDF, final predicted classes and intermediate predicted classes are highly consistent. Because RA has no effect on the predicted classes $p^0$, it indicates that the way for RA to improve robustness in testing is to guide models to extract robust feature during training.
>
> 2. Yes. And because predicted classes of $p^0$, $p^1$ and $p'$ in Tab. 1 in the PDF are highly consistent, it indicates that RA and the original block cooperate to keep the good representation and have a counteracting effect. If we remove RA, it will lead to an unbalance in the final layer, which may cause a decrease in robustness. Poor performance of Final probability vector without RA $p''$ has proven our opinion.
>
>
>
> **Minor Q1) What does it mean to freeze natural examples (in line 65)?**
>
> **A7)** It means we stop the back-propagation gradient of the feature of natural examples and only move other examples in the feature space. We explain it in lines 175-177.
>
> **Minor Q2) What does "selected carefully" mean in line 68?**
>
> **A8)** It means in adversarial contrastive learning, negative examples are often chosen from the dataset by a specific selection strategy. We point it out in lines 197-198.
>
> **Minor Q3) What are the problems mentioned in line 158?**
>
> **A9)** 1. How can we design proper formulae to calculate the negative contrast? 2. How can we obtain suitable examples as negative examples? We have explained them in lines 159-160.
>
> **L) The effectiveness on large datasets.**
>
> **A10)** We have improved our experiments on Tiny-ImageNet. As Tab. 2 in the PDF shows, our method has made great progress in robustness (31.44%\textgreater14.78%) compared with all the baselines and good improvement in clean accuracy(43.83%$\textgreater$38.61%), indicating its effectiveness on large datasets.

---

> > ### Author Response · Authors · 2023-08-14
> >
> > Dear Reviewer 1pqm,
> >
> > We hope that our responses can address your concerns and gain your increase in the rating score. If your have any new questions, we will response positively. Thanks for your time again.

---

> > ### Comment · Reviewer_1pqm · 2023-08-17
> > **Requires Further Improvement and Clarification**
> >
> > Dear authors,
> >
> > I want to acknowledge that I have read your response and appreciate your effort. However, I still believe that the contribution of the ANC component in your proposed method is quite limited, as indicated in Table 1, even though a significant portion of the paper is dedicated to this approach.
> >
> > While you have made attempts to address the questions I raised earlier, I strongly recommend that you carefully review the weaknesses highlighted in my reviews to further enhance your work.
> >
> > Best,

---

> > > ### Author Response · Authors · 2023-08-18
> > >
> > > **Writing Style.2: concerns about shared features by class 0 with other classes**
> > >
> > > In Figure 1 and Figure 2, We show the overall distances between natural examples and OEs rather than distances between pairs in two specific classes, because we want to obtain the overall distribution of feature distance to prove our opinions. To solve your concern about the common feature shared by class 0 with other classes, we fix class 1, class 3, and class 7 as natural samples and repeat the experiments respectively. The results are consistent with the previous results, indicating that our conclusions are sufficiently credible.
> > >
> > >
> > > **Content.1, 2: "well-trained DNN" and definition of an Adversarial Example**
> > >
> > > We follow your advice to revise sentences in L20-23 as follows.
> > > Given a naturally trained DNN and a natural example, an adversarial example can be generated by adding small perturbations to the natural example. Adversarial examples can always fool models to make incorrect output. At the same time, it is difficult to distinguish adversarial examples from natural examples by human eyes.
> > >
> > >
> > > **Content.3: the description of Figure 1**
> > >
> > > First, we will correct Gaussian distribution to bell-shaped distribution in L45~47 and line 46 and line 292.
> > > Second, we object to the view that the feature distance has been optimized well enough.
> > > 1. If the distances between natural and adversarial samples are considered to be sufficiently small, then their natural and robust accuracy should be quite close. But in fact, the natural accuracy is much higher than the robust accuracy (e.g., 80.90%>44.35% in Table 1). Though a cosine similarity between 0.9 and 0.99 seems a very small distance between natural examples and adversarial examples, their representations still need to be optimized to be more similar to reach similar accuracies.
> > > 2. If the distances between natural samples and negative samples are considered to be sufficiently large, then the distances of different classes are large enough for classification. But in fact, the robust accuracy is obviously not high enough (e.g., 44.35% in Table 1).
> > > 3. High natural accuracy does not mean that the class space is distant enough. It only shows the boundaries of different classes are divisible. But in the task of adversarial training, our targets are both natural accuracy and robust accuracy. The distances should be enough not only to distinguish natural samples of different classes, but also to keep the adversarial samples located at the class boundary separable enough. So it is necessary to maintain a large margin to correctly classify adversarial examples. Exclusion is proposed to achieve this goal.
> > >
> > >
> > > **Content.4: the definition of examples of other classes (OEs)**
> > >
> > > For the definition, we define OE in L43-44 and L148-149. OE is an example of other classes (with a different label from that of the natural example). In this paper, we choose OEs as negative examples, following adversarial contrastive learning. When training, we can choose selection strategies or our generation strategy to get OEs. To make it more clear, we have made Table 2 in the supplementary materials. And we will add some detailed definitions to Notations.
> > >
> > >
> > > **Content5: the meaning of 'partial parameters'**
> > >
> > > 'partial parameters' means those weights of the linear layer that are used to calculate the probability of the target class. Target class means label class when training, and denotes predicted class in the testing. We will add this description to Section 3.3 to help readers understand it.

---

> > > ### Author Response · Authors · 2023-08-18
> > >
> > > **Approach and Methodology.1(1): the Asymmetric negative contrast based on probabilities (ANC) does not make significant contributions**
> > >
> > > Firstly, ANC can obviously improve natural and robust accuracy on the basis of baseline (e.g., 1.8% in natural accuracy and 5.7% in robust accuracy against PGD). And Figure 3 has also shown that ANC does promote exclusion and keeps large margins between different classes.    So it's not objective enough to think ANC does not make significant contributions to the clean accuracy and robustness.
> > >
> > > Secondly, unlike adversarial examples located at the boundaries of classes, natural examples generally set in the center of class space. So classifying natural examples are easier than classifying adversarial examples. It causes natural classification is benefited from ANC more than adversarial classification. On the other hand, RA helps alignment and pull adversarial examples close to natural ones, which makes the classified accuracy of adversarial examples become close to the corresponding natural accuracy. Naturally, it contributes to robustness more.    It's not surprising that individual ANC does better in clean accuracy and RA does better in robust accuracy.
> > >
> > > Thirdly, compared with individual ANC and RA in Table 3, ANCRA has a 1% increase in natural accuracy, and a tiny increase in robustness against all the attacks except AutoAttack. And its robustness against AutoAttack is down by 1.3%. Though there is always a trade-off between natural accuracy and robustness, a lot of researchers are dedicated to maintaining good natural accuracy in AT, even at the expense of robust accuracy. So natural accuracy is also an important indicator for adversarial training. Overall, the combination of two techniques does well in the trade-off and obtains a good improvement in clean accuracy without sacrificing much robustness.
> > >
> > > Fourth, the motivation and design of ANC are enlightening and important contributions.  ANC has improved performance, which indicates our ideas of exclusion for robust feature are correct. We believe that not only the technique itself but also our motivation and ideas can bring new insights to the community.
> > >
> > >
> > > **Approach and Methodology.1(2): reverse attention (RA) is not studied well enough and lacks supporting experiments and theoretical insights**
> > >
> > > We have shown our motivation for RA in L68-72 on Page 2 and L213-220 on Page 6. We have shown its principle and theoretical insights in L235-257 on Pages 6-7. Reviewer PqGt thinks it is intuitive and has been properly justified in the manuscript. And experiment results in Fig. 3 on Page 7, Tab. 1 on Page 8, and Tab 2 on Page 9 have proven its effectiveness.
> > >
> > >
> > > **Approach and Methodology.2: the training process of ANC and the description of the last paragraph in the introduction**
> > >
> > > For the training process of ANC, we add a diagram to help readers understand each step of the implementation, which is shown in Fig. 1 in the PDF. And We use the detailed caption to describe the calculation process in these diagrams.
> > > For the description, we guess you mean PP and NP may cause readers hard to understand the contribution. So we have revised it as follows:
> > >
> > > To address the issue, we propose an AT framework to concentrate on robust representation with the guidance of the two characteristics. Specifically, we suggest two strategies to meet the characteristics, respectively. For exclusion, we propose an asymmetric negative contrast based on predicted probabilities, which freezes natural examples and pushes away OEs by reducing the confidence of the predicted class when predicted classes of natural examples and OEs are consistent. In particular, we find OEs generated by the targeted attack are more beneficial for correct classification than those selected carefully. For alignment, we use the reverse attention to weight feature by parameters of the linear classifier corresponding to target classes, which contains the importance of feature to target classes during classification. Because the feature of the same class gets the same weighting and feature of different classes is weighted disparately, natural examples and AEs become close to each other in the feature space. Empirical evaluations show that AT methods combined with our framework can greatly enhance robustness, which means the neglect of learning robust feature is one of the main reasons for the poor robust performance of AT. In a word, we propose a generic AT framework with the Asymmetric Negative Contrast and Reverse Attention (ANCRA), to learn robust representation and advance robustness. Our main contributions are summarized as follows:

---

> > > ### Author Response · Authors · 2023-08-18
> > >
> > >
> > > **Experiments.1: concerns about the false sense of security without ‘p’**
> > >
> > > Attack with "p" is the adaptive attack for our method, which is harder to defend against than general white box attacks. So it's normal to gain lowered robustness when we test by adaptive with "p". This situation happens a lot in other work. Notice that the performance of our method against the adaptive attack still surpasses that of other methods against normal white box attacks (e.g., 61.68% > 48.88%). It shows our method does make some success in robustness.
> > >
> > >
> > > **Experiment.2, 3: error bars and reproducibility**
> > >
> > > We have shown all the details of implementation in L266-273 on Page 7 and supplementary materials. We have made each experiment at least three times, and the variation of accuracy is generally less than 1.7%, which is marginal compared with the improvement. And we have provided Area Chairs with the anonymized link of our code.
> > >
> > >
> > > **Experiment.4: reproducibility for TRADES**
> > >
> > > From our experience, your code may be different from ours in learning rate, weight decay and step size of attacks. We set the learning rate as 0.01, set the weight decay as 0.0002 and set step size of attacks as 0.007, which we have written in lines 267, 268 and 272. The code is also available. Besides, Vanilla TRADES maximizes the KL loss to generate adversarial examples for training, but we maximize the CE loss to generate adversarial examples like PGD. By modifying these, we keep three baselines (PGD-AT, TRADES, MART) having the same method to generate adversarial examples for training.
> > >
> > >
> > > **Experiment.5: the results on Tiny-ImageNet**
> > >
> > > We have conducted some experiments to provide three baselines. As shown in Tab. 2 in the PDF, our methods made obvious progress in robustness compared with all the baselines, indicating its effectiveness on big datasets.

---

> > > ### Author Response · Authors · 2023-08-21
> > >
> > > We hope that our responses can address your concerns about weaknesses and gain your increase in the rating score. If you have any new questions, we will respond positively. Thank you for your suggestions. We will improve our paper and experiments.

---

### Official Review · Reviewer_K7R1 · 2023-07-07

**Soundness:** 3 good
**Presentation:** 3 good
**Contribution:** 3 good
**Rating:** 7
**Confidence:** 4

**Summary:**

In this paper, the authors address a notions of exclusion and alignment in representaion learning for robust adversarial training (AT). They propose a generic framework for AT that includes asymmetric negative contrast and reverse attention in order to obtain robust representation. In addition, they propose to weight feature by parameters of the linear classifier as the reverse attention, to obtain class-aware feature and pull close the feature of the same class. The authors show empirical evaluations on three benchmark datasets, showing the improved robustness under AT and as well as giving increased performance in comparison to state-of-the-art methods.

**Strengths:**

Originality. The proposed  AT framework that concentrates on robust representation with the guidance of the two characteristics like exclusion and alignment appears to be novel. Generating and using negative samples by targeted attack to assist learning also appears novel.

Quality. The quality is good. The paper is nicely motivated. The idea seems reasonable regarding the use two characteristics for robust representation: (1) exclusion: by pushing away the feature representation of natural examples from the feature representations for examples coming from other classes; and (2) alignment: by pulling close to each other the feature representation of the natural examples and the feature representation for the corresponding adversarial examples. The approach can be used in a plug-and-play manner for a number of defence methods, while the empirical validation shows advantages under the setting of white-box and adaptive attacks. While on table Table 2, we see consistent improvement for the used attacks with p, while we do not have results for comparison when we do not use p.

Clarity. The paper organisation, the presentation and the writhing is good. I find the paper easy to follow on most parts. It might be beneficial some parts of the paper to be explained more easily so that the main idea behind the paper to be more obvious. Fig 2. might be improved to more concretely pin point the cases of class confusion issue. In Sections 3.2.1, the section within the lines 162-169, a bit difficult to rad on the first pass regarding the cases covered by the class confusion issue. I would suggest an additional small figure to illustrate intuitively and more obviously the cases. In Sections 3.2.1, the section within the lines 170-172 is not clear (not sure when the some sentences end and begin). Section 3.2.2 could benefit further polishing. Maybe a table or small figure describing what is negative samples or pairs (i.e., OEs), natural negative examples/pairs, adversarial negative examples/pairs, positive examples/pairs, adversarial negative examples/pairs, etc. could be helpful. The text in the lines 234-257 could also benefit sharpening, since some explanations are not that smoot and straightforward at least to my understanding. The results on Table 4 are interesting, but maybe the table could be reorganised and more clearly shown so that we can have a better grasp of what is the actual advantage.

Significance. Adversarial training is important topic for privacy and security applications. This paper proposes an approach towards more reliable defence of adversarial attacks. The approach also seems to provide improvements on top of other approaches like the TRADES approach. Generating and using negative samples by targeted attack to assist learning seems valuable. Adversarial training with reverse attention on the feature level seems interesting.

**Weaknesses:**

Weaknesses
- In the empirical evaluation, some tables could be made more obvious (please see the comment on clarity for Table 4).
- The black box attacks not considered and comparison for black box attacks not done.



**Questions:**

Could the author comment about the use of the proposed approach under black box attacks?

Does the reverse attention works only for the TRADES approach?

I wonder of the asymmetric negative contrast based on probabilities is not somewhat artificially enforced, could the authors elaborate on that?

**Limitations:**

Limitations

- By generating negative samples by targeted attack, we use prior on about the possible attack, so possibly, the defence of the approach is biased towards the generated negative samples by the used targeted attack (or similar attacks).
- In the case of black box attacks when we do not have a prior about the attack, I'm not sure how much the proposed approach could help.

---

> ### Author Rebuttal · Authors · 2023-08-08
>
> We are grateful for your approval of the strengths and your constructive suggestions. The answers to your questions are as follows:
>
> **Clarity and W1) Some parts of the paper need a clear explanation to easily show our ideas.**
>
> **A1)** Thanks for your advice about clarity, we will add a dotted box to Fig. 2 on Page 5 to directly point out the practical instance of **class confusion**. Additionally, we will improve our text in lines 162-169, lines 170-173, and other places you and other reviewers have mentioned. Subject to word limitation, we can only show a part of them below.
>
> **Improved text in lines 170-173 on Page 4**: To alleviate the problem of class confusion, we should reasonably control the repulsion of negative contrast. We propose an asymmetric method of the negative contrast, Sim^α(x, xo), to decouple the repulsive force into two parts. It contains a one-side push from the natural example to the OE and a one-side push from the OE to the natural example, given by:
>
> For the **definition in Section 3.2.2**, we have defined adversarial examples (**AEs**) and examples of other classes (**OEs**) in lines 43-44 on Page 2 and lines 148-149 on Page 4. A natural example means one natural example used to train and test in the current batch. AE is the corresponding adversarial example of the natural example. OE is an example of other classes (with a different label from that of the natural example). In this paper, we choose AEs as **positive examples** and OEs as **negative examples**, following adversarial contrastive learning. And we don't define **negative samples** explicitly because they are a common concept in contrastive learning, which means natural examples with different labels from anchors. **Natural negative examples** are negative examples selected in the batch or dataset, and **adversarial negative examples** are made from natural negative examples by our strategy of targeted attack. These two concepts are the same as their literal meaning. We only define **negative pairs** and **positive pairs** in lines 63-64 on Page 2 and lines 135-137 on Page 4. Negative pairs are double-element sets of natural examples and their negative examples(i.e., OEs), and positive pairs are double-element sets of natural examples and their positive examples(i.e., AEs). To make it more clear, we have made Tab. 2 in the supplementary materials. And we will add some detailed definitions to Notations on Page 4 in the revised version.
>
> For **Tab. 4** on Page 9, we plan to add a time comparison table to highlight our strengths. As shown in the table below, our strategy costs less time than the average of these selection strategies (9.8 hours) but achieves the best performance. Besides, Soft-LS and Hard-LS pick up samples with specific predicted classes from other classes, which suffer from the risk of not finding suitable samples in a single batch. While our strategy only needs a random sample with a label different from the natural one, it is hardly possible for all the samples in a batch to have the same label. That's one of our strengths.
>
> |  Defense  | Random | Soft-LS | Hard-LS | Targeted attack |
> |  ----  | ----  | ----  | ----  | ----  |
> | Total time(hours) | 6.9(±0.2) | 11.4(±0.1) | 11.3(±0.3) | 9.3(±0.4) |
>
> **Q1, W2, L2) The effectiveness of our method against black box attacks.**
>
> **A2)** Our method can be trained to defend against black box attacks because we need labeled data only in the training rather than testing. We have made experiments against transfer-based black box attacks. We craft adversarial examples by PGD-100 on the source model and then attack target models. The results are shown in the table below. Notice that all the source models and target models are ResNet-18, so it is easy to be attacked. Our method gains the best robustness among all the methods, indicating the effectiveness of our method in the black box scenario.
>
> |  Target \ Source  | PGD-AT | TRADES | MART |
> |  ----  | ----  | ----  | ----  |
> | PGD-AT | 44.73 | 58.25 | 59.65 |
> | TRADES | 58.91 | 48.53 | 60.21 |
> | MART | 58.66 | 58.46 | 49.26 |
> | TRADES-ANCRA | 62.03 | 60.43 | 62.23 |
>
>
> **Q2) Does the reverse attention works only for the TRADES approach?**
>
> **A4)** No. As we have highlighted in line 81 on Page 3 and line 263 on Page 7, it can be combined in a plug-and-play manner with other defense methods. And it does not need any extra components. As shown in the experiments, it can be successfully combined with PGD-AT, TRADES and MART.
>
> **Q3) The reasonability of asymmetric negative contrast based on probabilities (ANC).**
>
> **A5)** From the view of logic and motivations, we notice that adversarial learning lacks robust feature learning. Thus, we propose two characteristics to guide it, which are exclusion and alignment. We propose two techniques for two characteristics, respectively. And we design ANC to promote exclusion. Therefore, it sounds logical and fits our motivations.
>
> From the view of the technique itself, experiments have proven that it does help exclusion and improves both generalization and robustness, which achieves our goal.
>
> **L1, L2) The prior learned from targeted attacks.**
>
> **A3)** When training, we craft negative examples via the targeted attack to teach models to learn prior knowledge of adversarial noise gradually. The prior knowledge can easily transfer from the training set to the test set, because we default that the data distribution of the training set and the test set is the same. We are not sure whether it has a bias against targeted attacks, but it helps models in robustness against various attacks from experiment results. It always works in the inference process under whatever kinds of attacks.

---

> > ### Author Response · Authors · 2023-08-14
> >
> > Dear Reviewer K7R1,
> >
> > We hope that our responses can address your concerns. If your have any new questions, we will response positively. Thanks for your time again.

---

> > ### Comment · Reviewer_K7R1 · 2023-08-21
> > **Response to the Rebuttal**
> >
> > Thanks to the authors for their feedback, it addresses some of the concerns that were raised. I also read the other reviews and the discussion there. I'm with the opinion to keep my initial score.

---

### Official Review · Reviewer_BjXk · 2023-07-26

**Soundness:** 3 good
**Presentation:** 4 excellent
**Contribution:** 4 excellent
**Rating:** 6
**Confidence:** 3

**Summary:**

This paper presents two characteristics of robust features, exclusion and alignment, and proposes a novel adversarial training method with asymmetric negative contrast and reverse attention. For exclusion, it introduces asymmetric negative contrast loss and generates adversarial negative examples by targeted attacks to push out examples of other classes in the feature space, and for alignment, it introduces reverse attention that weights features based on the parameters of the linear classifier to obtain class-aware features. Experimental results on three datasets show that the proposed method significantly improves the robustness of the models.

**Strengths:**

1. This paper introduces two characteristics that robust features should have: exclusion and alignment, and proposes a new AT framework that enables models to learn robust features effectively. It can be used in a plug-and-play manner and works well with existing AT methods.
2. Specifically, asymmetric negative contrast loss for exclusion of robust features, a technique to create hard negative samples through targeted attacks, and reverse attention using class information for alignment are proposed, which induce the characteristics of robust features. To my knowledge, these techniques are novel in AT.
3. The proposed method records the SoTA performance in experiments on CIFAR-10, -100, and Tiny-ImageNet. The performance improvement shown in Table 1 is impressive.
4. Overall, the paper is well written to help the readers understand the presented concepts. In detail, figures 1 and 3 show the statistical differences between NEs and OEs and NEs and AEs, making it easier for the reader to understand the difference in the distribution of features and the effectiveness of the proposed AT.  Figure 2 also helps to illustrate the problem of class confusion.


**Weaknesses:**

1. The practical implementation details for Reverse Attention in Section 3.3 are unclear. Looking at line 228 of the paper, it says that the proposed method uses p and p' together to train the model, but it is ambiguous in what way the proposed method uses them together (e.g., does it use p+p' or alternate between p and p'). Releasing the source code in the future will answer a lot of questions, but it would be nice if it could also be clarified in the paper.
2. The experimental results for "adaptive attack" in Section 4.3 raise the question of whether the experimental results in Section 4.2 are an unfair comparison. Therefore, it is necessary to clarify what the role of p is and whether this is the role of the key in the gray-box setting.
3. Table 1 in the supplementary material, the results on Tiny-ImageNet, does not have baselines, making it difficult to see performance gains.
4. Making hard negative examples via targeted attacks requires additional computation, but according to Table 4, the resulting performance gain is small. This paper also needs a clear and specific description of the negative sample generation (such as whether PGD-N was used).  In the supplementary material, it is described as if they used the AEs that are predicted as the classes of NEs in the current batch as OEs, which is difficult to apply in cases with a large number of classes like CIFAR-100 unless the batch size is large enough.


**Questions:**

(Typos)

In line 331, Trdeas-ANCRA -> Trades-ANCRA

In Table 2 in Supplementary material, Condation -> Condition

**Limitations:**

In this paper, possible limitations are not discussed separately.

---

> ### Author Rebuttal · Authors · 2023-08-08
>
> We are grateful for your approval of the strengths and your constructive suggestions. The answers to your questions are as follows:
>
> **W1) The unclear computing details for Reverse Attention (RA).**
>
> **A1)** Subject to page limitation, we focus on the principle and function of our method in the paper. It seems to lack practical implementation details for RA. We will provide a description and diagram to show how RA works in the supplementary materials in the revised version. The diagram is shown in Fig. 2 in the PDF. And the description is as follows: Firstly, we obtain two auxiliary probability vectors ($p^0$ and $p^1$) in the two blocks in the final layer of the model. Secondly, we calculate auxiliary loss $l$ and get predicted classes $h(x)$ to choose corresponding linear parameters $\Omega^j$ for testing. After multiplying the feature and $\Omega^j$, we get the final output $p'$ to calculate loss $l'$. Finally, we add $l$ and $l'$ by a specific weight to get the total loss and then backward.
>
> **W2) Concern about the fairness of the comparison experiments in Section 4.2.**
>
> **A2)** To prove its fairness, we first introduce the detailed implementation of the experiments. Our experiments in Section 4.2 are conducted on four robust models trained with four defense methods. First, we sample clean data of class 0 from the test set, and generate adversarial examples via untargeted PGD-10. Second, we pick up clean examples from other classes, and make them into negative examples by our generation strategy via targeted PGD-10. And then we measure the corresponding distance indicators. The processing procedure is completely consistent between four trained models and there is no unfairness in the whole procedure.
>
> We are not so sure whether your concern about the unfair comparison derives from the method of attacks. We choose PGD as a representative of common white box attacks, to gain results of feature distributions that can better show how this defense performs against general attacks rather than the adaptive attack. Because the adaptive attack with auxiliary probability vector $p$ is specially designed for our method, the defense results against the adaptive attack cannot represent the performance of the model in general defense scenarios. The generalization of defense results is what we want. Therefore, it's fair to choose white box PGD rather than adaptive PGD to attack our method.
>
> **W3) The lack of baselines on Tiny-ImageNet.**
>
> **A3)** We have conducted some experiments to provide three baselines. As shown in the table below, our methods made obvious progress in robustness compared with all the baselines, indicating its effectiveness on big datasets.
>
> |  Defense   | PGD-AT  | TRADES  | MART  | PGD-AT-ANCRA  | TRADES-ANCRA  | MART-ANCRA  |
> |  ----  | ----  | ----  | ----  | ----  | ----  | ----  |
> | Nat | 41.31(±1.2) | 37.27(±0.5) | 38.61(±0.9) | 43.02(±1.7) | 38.94(±0.6) | 43.83(±0.9)  |
> | PGD  | 10.28(±0.7) | 16.30(±0.8) | 14.78(±0.5)  | 29.79(±0.7)  | 31.24(±1.4) | 31.44(±0.4) |
>
> **W4) The strengths and description of our strategy of negative sample generation.**
>
> **A4)** Though its performance gain is not significant, it has four strengths.
> 1. The strategy of negative samples via targeted attack gains the best performance over other selection strategies, especially in the last epoch.
> 2. As shown in the table below, our strategy costs less time than Soft-LS and Hard-LS by 2 hours and more time than Random by 2 hours. It means it spends less time than the average of these selection strategies but achieves the best performance, which is attractive and excellent.
> 3. And as you have mentioned, Soft-LS and Hard-LS pick up samples with specific predicted classes from other classes, which suffer from the risk of not finding suitable samples in a single batch. While our strategy only needs a random sample with a label different from the natural one.
> 4. Though it learns a prior about adversarial noise only from targeted attacks, the prior still works when defending against other kinds of attack.
>
> |  Defense  | Random | Soft-LS | Hard-LS | Targeted attack |
> |  ----  | ----  | ----  | ----  | ----  |
> | Total time(hours) | 6.9(±0.2) | 11.4(±0.1) | 11.3(±0.3) | 9.3(±0.4) |
>
>  For the description of the negative sample generation, we have written in lines 197-201 on Page 5 and in Tab. 2 in the supplementary materials. Firstly, we gain a batch of natural examples as natural examples, whose labels are target classes. Secondly, for each natural example, we randomly choose an example with a label different from the target class from this batch, named by the natural negative example for easy understanding. Thirdly, we attack these natural negative examples from original classes toward target classes by targeted PGD-10, which is written in L201 on Page 5.
>
>
> **Q) Typos.**
>
> **A5)** We are sorry for typos and other writing errors in our paper. We will follow your and other reviewers' suggestions to fix all the typos as well as other writing errors. We will do our best to improve this issue in the revised version.
>
> **L) The missing limitations in our paper.**
>
> **A6)** We will discuss limitations in the revised version. Details are as follows. Though reverse attention contributes to both generalization and robustness, its accuracy shows high dependence on intermediate predicted classes. It may become cause degraded performance when faced with powerful attacks. We will further study it and hope to improve its robustness in the future.

---

> > ### Author Response · Authors · 2023-08-14
> >
> > Dear Reviewer BjXk,
> >
> > We hope that our responses can address your concerns and gain your increase in the rating score. If your have any new questions, we will response positively. Thanks for your time again.

---

> > ### Comment · Reviewer_BjXk · 2023-08-20
> >
> > I would like to thank the authors for their effort in preparing the detailed and kind rebuttal.
> > The computational details explained by the authors in their rebuttal are very helpful for my understanding, and the experimental results of the baselines on Tiny-ImageNet demonstrate  the performance improvement of the proposed method.
> >
> > Regarding W2, the concern I raised was that it might be unfair to compare primarily on the basis of robustness against adversarial attacks from attackers who don't know 'p'. Since many adversarial defense strategies have been bypassed by white-box adaptive attacks, I believe that the proposed method should also be ‘primarily’ evaluated by its robustness against white-box adaptive attacks. In the future, it would be nice to include adaptive AutoAttack results when p is known.
> >
> > Regarding W4, as we need to see and evaluate the performance of both the best model and the last epoch model, the performance improvement from the strategy of generating negative examples through targeted attacks seems quite limited. The differences in experimental settings also make it difficult to compare the robustness of SoTA models in RobustBench with the robustness of the experimental results in this paper.
> >
> > Although more experimental results are needed to show a clear performance improvement, I think the motivation behind the proposed methods is novel, so I would maintain or slightly downgrade my initial rating.

---

> > > ### Author Response · Authors · 2023-08-21
> > >
> > > About **W2**, we have conducted new experiments on WideResNet-28-10 and WideResNet-34-10. As shown in the Table below, our method has made great enhancements in robust accuracy. The accuracies of our method against adaptive AutoAttack are even higher than those of baselines against AutoAttack (**51.99**%>**50.79**% and **51.85**%>**51.11**%). This indicates its effectiveness against adaptive attacks.
> > >
> > > | Model            | Method       | NAT    | AutoAttack | Adaptive AutoAttack |
> > > |------------------|--------------|--------------|------------|---------------------|
> > > | WideResNet-34-10 | TRADES       | 82.04 | 50.79      | \                   |
> > > | WideResNet-34-10 | TRADES-ANCRA | 83.19 | 66.28      | 51.99               |
> > > | WideResNet-28-10 | TRADES       | 82.47 | 51.11      | \                   |
> > > | WideResNet-28-10 | TRADES-ANCRA | 83.61 | 66.08      | 51.85               |
> > >
> > > About **W4**, as we have mentioned in the global response, our generation strategy via targeted attacks has four strengths in different aspects. Though it doesn’t show a large improvement in the current settings, the idea of it may contribute to other work such as Adversarial Contrastive Learning. And we will provide more experiments to show its effectiveness in the revised version, such as fair contrastive experiments with SOTA methods in RobustBench.
> > >
> > > Thank you for your recognition and suggestions. We will improve our paper and experiments in the revised version. Thanks for your time again!

---

### Official Review · Reviewer_PqGt · 2023-07-26

**Soundness:** 3 good
**Presentation:** 2 fair
**Contribution:** 3 good
**Rating:** 5
**Confidence:** 3

**Summary:**

This work aims to improve the adversarial training (AT) techniques from the perspective of learning robust representation representations. Specifically, the authors highlight two characteristics of having robust features. Exclusion: the similarity of features of samples of one class should be very less from the features of samples of other classes, so that model can differentiate between features of different classes for better classification.  The second attribute is Alignment: the gap between features of adversaries and clean samples of same class must be very small, which would increase model's robustness against perturbed samples.

To effectively satisfy these conditions, this work proposes two techniques, (1) to enforce exclusion, a asymmetric negative contrast loss is proposed which minimizes the clean sample similarity with negative samples of other classes, crafted by the adversarial attack. (2) to satisfy alignment: reverse attention strategy is proposed which align together the features of training examples which belong to the same class.

The proposed method is compatible with existing adversarial training techniques. Extensive experiments are conducted where the proposed method improves the natural accuracy as well as the robust accuracy when combined with existing AT algorithms.

**Strengths:**

Strengths:
1) The idea of improving adversarial robustness of model by explicitly learning robust representations seems interesting. Although the traditional AT and contrastive learning based AT implicitly learns robust features, this method attempts to achieve the same in more explicit manner.

2) The proposed techniques of utilizing asymmetric negative contrast loss and reverse attention to achieve exclusion and alignment during adversarial training are intuitive and are properly justified in the manuscript.

3) The method provides impressive results as compared to previous state-of-the-art approaches.



**Weaknesses:**

Weaknesses:

1) There are concerns regarding the proposed reverse attention strategy. During the testing, the true labels are not known and reverse attention uses the predicted label h(x) to calculate z'. In case the model provides wrong predicted class label, the corresponding z' will be also then multiplied with wrong classifier vector. This will further lead to degraded performance. The authors have not tried to address this scenario.
2) From the works of [1] and [34], how is the proposed reverse attention different? Unfortunately the authors have not provided any comparisons or contrast.
3) In the ablation studies provided in Table 3, combining ANC and RT marginally improves results as compared to individual ANC and RT results. It looks like there is some sort of competition between the both proposed techniques.
4) Similarly, the use of negative samples via proposed targeted attack in Table 4 shows marginal improvements overall.
5) The paper is very difficult to understand, especially for the readers who are new to the technique of adversarial training. The presentation can be significantly improved.

Minor weaknesses:
typo at line 281 scenaios -> scenarios

**Questions:**

Please see weaknesses section.

**Limitations:**

The authors have not discussed any limitations of their work.

---

> ### Author Rebuttal · Authors · 2023-08-08
>
> We are grateful for your approval of the strengths and your constructive suggestions. The answers to your questions are as follows:
>
>
> **W1) The wrong predicted class leads to the wrong weighted feature and degraded performance.**
>
>
> **A1)** The wrong predicted class always causes misclassification, but it does not significantly affect performance. First, the accuracies of predicted labels in the different blocks of reverse attention are shown in the table below. It shows that the final predicted results and intermediate predicted labels remain highly consistent. The dependence on intermediate predicted labels is a limitation of our method. We will add it to the revised version. Details are as follows. Though reverse attention contributes to both generalization and robustness, its accuracy shows high dependence on intermediate predicted classes. It may become cause degraded performance when faced with powerful attacks. We will further study it and hope to improve its robustness in the future.
>
> |     | Nat  | PGD | Adaptive PGD |
> |  ----  | ----  | ----  | ----  |
> | Predicted labels (first block) | 81.81 | 83.52 | 62.25 |
> | Predicted labels (second block) | 81.81 | 83.49 | 62.23 |
> | Final predicted labels  | 81.81 | 83.47 | 62.24 |
>
> Second, as shown in Tab. 2 on Page 9, our defense against the adaptive attack can still keep the best performance compared with all the approaches in Tab. 1 on Page 8 (e.g., 61.68%＞48.88% against PGD), indicating that this issue does not significantly affect performance. It is because reverse attention not only helps defense models in terms of weighting feature in the final layer, but also teaches models to extract robust feature in the whole process. So it still works in improving robustness faced with this problem.
>
>
> **W2) Difference of reverse attention compared with methods in [1] and [34].**
>
> **A2)** [1] propose an additional linear layer to learn which feature channels are important for classification. And they aim to gain its parameter to weight feature. Our reverse attention not only has achieved this target without any extra components, but also can help models learn robust feature in the whole process. And our method can be explained by alignment of robust representation. [34] do similar work with [1] and makes contributions on how to weight feature properly, which is orthogonal to our contributions. We have made a brief introduction to [1] and [34] in lines 114-119 in Related work on Page 3, where we point out that they rely on extra model components and do not explain the reason.
>
> [1] Bai, Yang, et al. "Improving adversarial robustness via channel-wise activation suppressing." arXiv preprint arXiv:2103.08307 (2021).
>
> [34] Yan, Hanshu, et al. "Cifs: Improving adversarial robustness of cnns via channel-wise importance-based feature selection." International Conference on Machine Learning. PMLR, 2021.
>
> **W3) The combination of the Asymmetric Negative Contrast (ANC) and Reverse Attention (RA).**
>
> **A3)** The combination of ANC and RA does well in the trade-off between generalization and robustness. Though there is a trade-off between natural accuracy and robustness, a lot of researchers are dedicated to maintaining good natural accuracy in adversarial training, even at the expense of robust accuracy. It indicates natural accuracy is an important indicator for adversarial training. As we can see in Tab. 3 on Page 9, ANC contributes to natural accuracy more than RA and  RA does better in boosting robustness. Compared with individual ANC and RA, ANCRA has a 1% increase in natural accuracy, and a marginal increase in robustness against all the attacks except AutoAttack. And its robustness against AutoAttack is down by 1.3%. The combination of  ANC and  RA obtains an excellent improvement in clean accuracy (81.70%) without sacrificing much robustness, which is a good result.
>
> **W4) The effectiveness of our strategy of negative samples generated by targeted attack.**
>
> **A4)** Here are four strengths.
> 1. Our strategy gains the best performance over other selection strategies, especially in the last epoch.
> 2. As shown in the table below, our strategy costs less time than Soft-LS and Hard-LS by 2 hours and more time than Random by 2 hours. It means it spends less time than the average of these selection strategies but achieves the best performance, which is excellent.
> 3. Soft-LS and Hard-LS pick up samples with specific predicted labels from other classes, which suffers from the risk of not finding suitable samples in a single batch. In experiments on CIFAR-100, they cannot find proper samples in 40%-70% of selections. While our strategy only needs a random sample with a label different from the natural one, it is hardly possible for all the samples in a batch to have the same label.
> 4. Though it learns a prior about adversarial noise only from targeted attacks, the prior still works when defending against other kinds of attack.
>
> |  Defense  | Random | Soft-LS | Hard-LS | Targeted attack |
> |  ----  | ----  | ----  | ----  | ----  |
> | Total time(hours) | 6.9(±0.2) | 11.4(±0.1) | 11.3(±0.3) | 9.3(±0.4) |
>
>
> **W5) The paper is difficult to understand for readers new to adversarial training.**
>
> **A5)** Subject to page limitation, we focus on the principle and function of our method in the paper. Some details and prior knowledge may be too difficult to understand by readers new to adversarial training. We add diagrams for individual techniques to help readers understand each step of the implementation, which is shown in Fig. 1 and Fig. 2 in the PDF. And We use the detailed caption to describe the calculation process in these diagrams.

---

> > ### Author Response · Authors · 2023-08-14
> >
> > Dear Reviewer PqGt,
> >
> > We hope that our responses can address your concerns and gain your increase in the rating score. If your have any new questions, we will response positively. Thanks for your time again.

---

> > > ### Comment · Reviewer_PqGt · 2023-08-15
> > >
> > > Dear Authors,
> > >
> > > Thank you for providing a rebuttal response and most of my concerns have been addressed. Its highly recommended to revise paper as discussed above and improve the overall presentation style so it can be clearer for the readers.
> > >
> > >
> > > I will consider to keep my current score.

---

> > > > ### Author Response · Authors · 2023-08-17
> > > >
> > > > Dear Reviewer PqGt,
> > > >
> > > > Thank you for your recognition and suggestions, we will improve our presentation style in the revised paper. Thanks for your time again and best wishes!

---

### Author Rebuttal · Authors · 2023-08-08

Dear **ALL** reviwers,

We are very grateful for your time and constructive suggestions. Here, we first summarize the **strengths** acknowledged by multiple reviewers.

We are encouraged by the approval of Reviewer PqGt, Reviewer BjXk, Reviewer K7R1 and Reviewer 2AVw for our **inspirable motivations**. They think our motivations and insights for learning robust feature are novel and inspirable. Besides, all of our reviewers agree that we have proposed **interesting techniques**. Reviewer PqGt, Reviewer BjXk and Reviewer K7R1 praise our techniques for their novelty. Reviewer PqGt thinks they are intuitive and are properly justified in the manuscript. Reviewer K7R1 thinks they are valuable. What's more, **impressive improvements in experiments** are affirmed by Reviewer PqGt, Reviewer BjXk, Reviewer K7R1 and Reviewer 2AVw. We have reached the best performance in robustness on different datasets and models, while we also make good improvements in clean accuracy.

Here, we answer the **common concerns** of several reviewers and then state the limitations of our work.

**Question 1: Whether the asymmetric negative contrast based on probabilities (ANC) is necessary and effective in our method? (Reviewer PqGt, Reviewer K7R1, Reviewer 1pqm)**

**A1**: Yes. From the view of logic and motivations, ANC is necessary for exclusion and robust feature learning. And from the view of empirical evaluations, our method benefits from ANC in terms of the trade-off between generalization and robustness. As shown in Tab. 3 on Page 9, ANCRA is the only method to have natural accuracy over 81.0%, and it still keeps similar robustness as individual reverse attention.

**Question 2: What are the strengths of our generation strategy of negative examples via the targeted attack? ( Reviewer PqGt, Reviewer BjXk, Reviewer K7R1)**

**A2**: **First**, our strategy gains the best performance over selection strategies, especially in the last epoch. **Second**, as shown in Tab. 3 in the PDF, our strategy costs less time than the average of selection strategies and achieves the best performance, which is excellent. **Third**, Soft-LS and Hard-LS need to pick up samples with specific predicted labels from other classes, which suffers from the risk of not finding suitable samples in a single batch. While our strategy only needs a random sample with a label different from the natural one. **Forth**, though it learns a prior about adversarial noise only from targeted attacks, the prior still works when defending against other kinds of attack.

**Question 3: What will happen if the wrong predicted class $h(x)$ leads to the wrong weighting in reverse attention (RA)? And how does RA work in testing? (Reviewer PqGt, Reviewer BjXk, Reviewer 1pqm)**

**A3**: 1. The wrong predicted class always causes misclassification, but this situation does not significantly affect performance. As shown in Tab. 1 in the PDF, the final predicted class and predicted class $h(x)$ remain highly consistent. However, as shown in Tab. 2 on Page 9 and Tab. 1 on Page 8, our defense against adaptive attacks keeps the best performance compared with all the approaches against white box attacks, indicating that the wrong prediction does not significantly affect performance in practical testing.

2. RA works by teaching models **extract robust feature** during training and having a **counteracting effect with the feature block**. Because RA has no direct effect on the high accuracy of predicted classes $p^0$ in Tab. 1 in the PDF, it indicates that RA guides models to extract robust feature during training. Because robust feature is with good generalization, it still works during testing.
Besides, as shown by the high consistency of $p^0$, $p^1$ and $p'$ in Tab. 1 in the PDF, RA and the original block cooperate to remain good representation in different blocks and have a counteracting effect. Poor performance of Final probability vector without RA $p''$ has proven our opinion.

**Limitations:**

**A4**: We will add limitations in the revised version. Details are as follows. Though reverse attention contributes to both generalization and robustness, its accuracy shows high dependence on intermediate predicted classes. It may become cause degraded performance when faced with powerful attacks. We will further study it and hope to improve its robustness in the future.

Moreover, we will follow suggestions to fix **typos** and writing errors in the revised version. We will improve vague sentences and add diagrams for individual techniques (e.g., Fig. 1 and Fig. 2 in the PDF) for easy understanding. Our **code** has been submitted to Area Chairs.

---

### Decision · Program_Chairs · 2023-09-21

**Decision:**

Reject

**Comment:**

The work seeks to improve Adversarial Training to learn robust features.

Reviewers and AC appreciate the overall motivation and the hard work authors have put into detailed responses. However, there were several concerns raised on the experimental evaluation, as well as some of the key limitations of the approach.

On the balance, AC feels the paper has value in terms of identifying the limitation of current AT approaches for learning robust feature representations but needs clear experimental results included on common architectures i.e., WideResNet models with direct comparisons on comparable approaches on RobustBench (as authors have indicated the initial experiments can benefit from careful design choices), proper comparisons with baselines on TinyImageNet and compute cost limitation in the main paper. The contribution of ANC component is also weak. A discussion and analysis on limitations e.g., the effect of wrong predictions on the final performance (esp. in hard datasets like TinyImageNet) will be helpful. The writing quality and typos also need to be fixed.

Although the paper is on the fence, AC feels the paper can be improved by considering reviewer comments carefully and improving further based on them. Therefore, the recommendation is that the paper cannot be accepted in its current form. The decision was discussed and endorsed by the SAC.